# A DARPin promotes faster onset of botulinum neurotoxin A1 action

Oneda Leka[1,4], Yufan Wu[1,4], Giulia Zanetti [2], Sven Furler[3], Thomas Reinberg[3], Joana Marinho[3], Jonas V. Schaefer[3], Andreas Plückthun [3], Xiaodan Li [1], Marco Pirazzini [2] & Richard A. Kammerer [1] ✉

In this study, we characterize Designed Ankyrin Repeat Proteins (DARPins) as investigative tools to probe botulinum neurotoxin A1 (BoNT/A1) structure and function. We identify DARPin-F5 that completely blocks SNAP25 substrate cleavage by BoNT/A1 in vitro. X-ray crystallography reveals that DARPin-F5 inhibits BoNT/A1 activity by interacting with a substrate-binding region between the α- and β-exosite. This DARPin does not block substrate cleavage of BoNT/A3, indicating that DARPin-F5 is a subtype-specific inhibitor. BoNT/A1 Glu-171 plays a critical role in the interaction with DARPin-F5 and its mutation to Asp, the residue found in BoNT/A3, results in a loss of inhibition of substrate cleavage. In contrast to the in vitro results, DARPin-F5 promotes faster substrate cleavage of BoNT/A1 in primary neurons and muscle tissue by increasing toxin translocation. Our findings could have important implications for the application of BoNT/A1 in therapeutic areas requiring faster onset of toxin action combined with long persistence.

Botulinum neurotoxins (BoNTs) produced by anaerobic bacteria of the genus *Clostridium* are the most poisonous bacterial protein toxins known[1–4]. BoNT intoxication in vertebrates causes botulism, a potentially life-threatening neuroparalytic syndrome[5]. Therefore, the toxins represent potential biological weapons. Despite their toxicity and as a result of their characteristics, including biological effectiveness and long persistence of action in patients, BoNTs are nowadays amongst the most widely used therapeutic proteins in various human neurological and non-neurological disorders[6,7]. Furthermore, they are used in cosmetic applications. Traditionally, BoNTs are classified as seven serologically distinct proteins, referred to as BoNT/A through BoNT/G, however, their genetic variability is further increased by the existence of more than 40 subtypes within serotypes A, B, E, and F[8,9]. Although many subtypes have not yet been well characterized, it was shown that some differ from the other proteins within a serotype with respect to their catalytic properties, substrate specificity, duration of action, and efficiency to enter neuronal cells[10–13]. Furthermore, there exist mosaic toxins between serotypes C and D[14] and partially between A and F[15]. Because they could outperform conventionally used BoNTs with respect to their biological activities, it appears important to characterize all BoNT subtypes. Recently, several BoNT-like sequences have been discovered in single bacterial strains whose occurrence is of great concern, because it indicates that *Clostridium botulinum* is able to horizontally transfer BoNT genes to other bacterial species[16–19].

BoNTs are produced as single precursors that share a common domain organization[20–23]. They are processed into mature toxins by cleavage into a ~50 kDa light chain (LC) and a ~100 kDa heavy chain (HC) that remain connected through an interchain disulfide bond.

The LC is a zinc-dependent endopeptidase that specifically cleaves members of the soluble N-ethylmaleimide-sensitive-factor attachment receptor (SNARE) family of proteins, which are key components of the vesicular fusion machinery within presynaptic nerve terminals. BoNT/A and E cleave synaptosomal-associated protein 25 (SNAP25), and BoNT/B, D, F, and G cut vesicle-associated membrane protein (VAMP). BoNT/C can cleave two substrates, SNAP25 and syntaxin[1,24,25]. SNARE cleavage blocks acetylcholine release at the neuromuscular junction, leading to a flaccid paralysis of muscles[5].

[1]Laboratory of Biomolecular Research, Division of Biology, Paul Scherrer Institut, 5232 Villigen PSI, Switzerland. [2]Department of Biomedical Sciences, University of Padova, 35121 Padova, Italy. [3]Department of Biochemistry, University of Zurich, 8057 Zurich, Switzerland. [4]These authors contributed equally: Oneda Leka, Yufan Wu. ✉e-mail: Richard.Kammerer@psi.ch

The HC consists of a 50 kDa N-terminal translocation domain ($H_N$) and a C-terminal receptor-binding domain ($H_C$) of similar size. A dual interaction mode involving two different receptors seems to be required for the neuro-specific binding of BoNTs[26]. Most BoNTs bind a polysialoganglioside (PSG) and synaptotagmin (Syt) or synaptic vesicle glycoprotein 2 (SV2)[1,3,27].

Upon binding to neuronal cell receptors, BoNTs are endocytosed into synaptic vesicles[1]. The low pH and the lipid environment within the synaptic vesicle are believed to initiate conformational changes in $H_N$[4,28]. Amongst those, exposure of a hydrophobic peptide through a viral-fusion-peptide-like pH-dependent molecular switch seems to initiate the first step of membrane insertion of BoNT/A1[29]. Subsequent conformational changes are thought to lead to the formation of an ion-conductive transmembrane channel through which the LC is translocated across the membrane. Once inside the reducing intracellular environment of the cytosol, the disulfide bond is reduced, resulting in release of the protease.

The existence of multiple BoNT and BoNT-like toxin molecules represents a great public health threat, because they can potentially all cause botulism or be engaged as bioweapons. As result of this public health concern, multiple strategies have been developed for the treatment and prevention of botulism[30,31]. Like the toxoid-based tetanus vaccination against the structurally related Tetanus neurotoxin (TeNT), vaccination using BoNT toxoid efficiently protects against botulism[30]. Although different next-generation vaccines are being developed to replace the discontinued toxoid vaccine[32], vaccination is seldom used because botulism is a rare disease with less than 200 cases reported per year in the United States[30]. More importantly, vaccination would have a negative impact on the therapeutic benefits of BoNTs for medical applications. As a result, passive immunotherapy using polyclonal antibody preparations and sera, individual monoclonal antibodies, and monoclonal antibody combinations are used for the medical treatment of botulism[30]. In addition, alternative strategies such as the application of camelid single-chain antibodies are also being developed[33–38].

A class of proteins whose usefulness as BoNT-inhibiting agents has not yet been explored are the DARPins (Designed Ankyrin Repeat Proteins)[39]. DARPins are small, engineered antibody mimetics derived from synthetic libraries of consensus ankyrin proteins that are very specific and show very high affinity to the target protein. They are successfully used in various research, diagnostic, and therapeutic applications. Considerable benefits of DARPins are their small size of 14 or 18 kDa (4 or 5 ankyrin repeats of which two or three carry a randomized surface and are flanked by N-capping and C-capping repeats) and their high stability and solubility. In addition, DARPins can be generated rapidly and cost-efficiently in a soluble form and high yields in the cytoplasm of *E. coli*.

Biomolecules like nanobodies and DARPins might contribute significantly to the development and of next-generation biomolecular antidotes against botulism. However, since they typically recognize conformational epitopes, nanobodies, and DARPins also represent invaluable tools to probe the structure and function of BoNTs.

In this work, we investigate the impact of DARPins selected against LC/A1 on toxin activity using biochemical and biophysical methods, X-ray crystallography and functional assays performed in vitro and in cells and muscle tissue.

## Results
### Generation and characterization of DARPins selected against LC/A1
DARPins were selected over four rounds of ribosome display[40–42] using a semi-automated 96-well platform against full-length LC/A1 (amino-acid residues Pro-2 to Lys-448) (see Materials and Methods for details). In total for each selection 380 single clones were screened using a 384-well-based Homogenous Time Resolved Fluorescence (HTRF) assay.

From the initial hits, 32 were sequenced. For the selection against full-length LC/A1, 25 specific and unique clones were identified. These were expressed and IMAC purified. Binding was confirmed using ELISA.

### DARPin-F5 inhibits the catalytic activity of LC/A1 in vitro
Because its C-terminus contributes to catalysis, the full-length LC represents the relevant target for BoNT/A1 inhibitor studies[35,43,44]. In a first step, we were searching for DARPins that inhibit the protease activity of full-length LC/A1. To this end, we incubated each of the 25 selected DARPins individually at different ratios with LC/A1 and monitored the cleavage of the recombinant human SNAP25 substrate peptide spanning amino-acid residues Met-141 to Gly-204. We identified six DARPins that completely or partially inhibited the proteolytic activity of BoNT-LC/A1. Among the candidates, DARPin 008-829-2386-F5 (named DARPin-F5 for short, Supplementary Fig. S1) was able to completely inhibit substrate cleavage at a DARPin/toxin molar ratio of 2.5:1 (Fig. 1a, Supplementary Fig. S7) while the remaining five DARPins showed partial inhibition of LC/A1 enzymatic activity to varying degrees.

Because the C-terminus of the catalytic domain has been implicated in enzyme/substrate interactions[35,44], we also tested the ability of DARPin-F5 to prevent the cleavage of the SNAP25 peptide substrate using two C-terminally truncated LC/A1 variants (residues Pro-2 to Gly-421 and Pro-2 to Gly-433). DARPin-F5 also completely inhibited proteolytic activity of both truncated LC/A1 variants, demonstrating that DARPin-F5 doesn't bind to the C-terminus of the catalytic domain.

In full-length BoNT/A1 the catalytic domain is surrounded by the belt region, which is considered a pseudo-substrate that prevents LC/A1 protease activity before its delivery into the neuronal cytosol[45]. Although it is unknown if LC and HC remain bound to each other after reduction of the connecting disulfide bridge, reduced full-length BoNT/A1 efficiently cleaves SNAP25. As observed for LC/A1, DARPin-F5 was found to also completely inhibit reduced full-length BoNT/A1 (Fig. 1b).

### Crystal structure of the LC/A1-DARPin-F5 complex
To understand at the molecular level how DARPin-F5 inhibits the catalytic activity of LC/A1, we next focused on determining the crystal structure of the complex. Attempts to crystallize DARPin-F5 with the full-length catalytic domain or a slightly shorter LC/A1 variant (amino-acid residues Pro-2 to Gly-433) were not successful, probably because these variants are too flexible for crystallization and prone to aggregation[43,46]. Therefore, we focused on an even more truncated LC/A1 variant spanning amino-acid residues Pro-2 to Gly-421, which was previously used to solve the crystal structure of a LC/A1-SNAP25 toxin/substrate complex[47]. Crystals suitable for structural work were obtained within 1–2 weeks and the structure of the LC/A1-DARPin-F5 complex was solved at a resolution of 2.5 Å (Fig. 2a and Supplementary Table S1; PDB code 8HKH). The crystal structure contains two toxin/DARPin complexes per asymmetric unit (rmsd value of 0.2 Å for 510 Cα atoms of both AC and BD complexes). In the following, we analyzed the AC complex.

SNAP25 wraps around the catalytic domain in a manner very similar to the belt[47]. The binding interface includes the α-exosite and the β-exosite, two major structural motifs that bind the ~60 residue-long substrate remote from the catalytic site. It has been shown that the exosites play important roles for substrate binding and specificity[47]. The crystal structure of the complex reveals that DARPin-F5 binds to a substrate-binding site between the α- and β-exosites of LC/A1, far away from the active site of the catalytic domain. The DARPin therefore inhibits SNAP25 hydrolysis by preventing substrate binding. Consistent with this observation, it has been shown that nanobodies preventing the binding of the substrate also show potent inhibition of LC/A1[35,36]. All BoNT-LCs share a globular domain fold in which a conserved substrate-binding groove extends from the

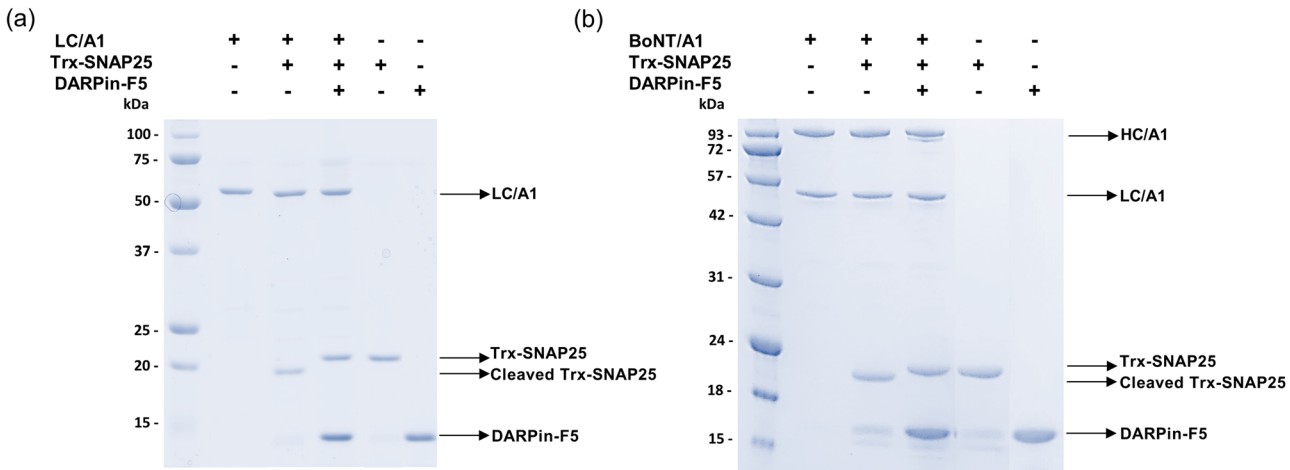

**Fig. 1 | Inhibition of enzymatic activity of BoNT/A1 and recombinant LC/A1 by DARPin-F5. a** LC/A1 was incubated with DARPin-F5 (DARPin/toxin ratio of 2.5:1). After incubation, the recombinant substrate Trx-SNAP25 was added to the mixture and the catalytic activity of LC/A1 was abolished (lane 4). **b** The same experimental conditions were applied to reduced full-length BoNT/A1. In both experiments, samples were analyzed by SDS-PAGE and Coomassie Blue staining. The position of

the BoNT/A1 heavy chain (HC/A1), BoNT/A1 light chain (LC/A1), recombinant full-length LC/A1 (residues P2-K448), cleaved and uncleaved Trx-SNAP25 substrate and DARPin-F5 are indicated by arrows. The positions of marker proteins are indicated. Experiments in (**a**) and (**b**) were repeated twice independently and yielded similar results.

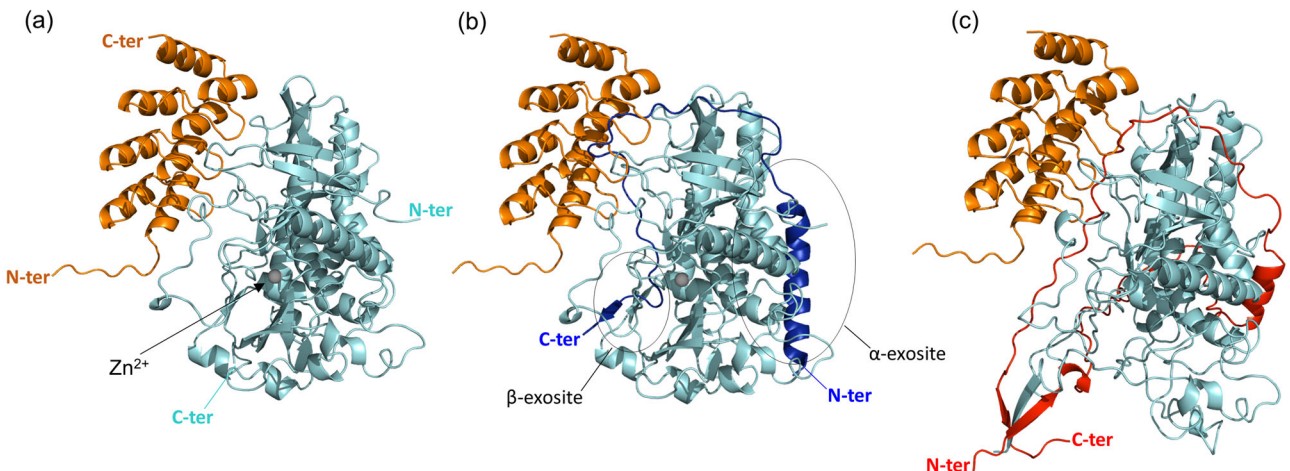

**Fig. 2 | Crystal structure of the LC/A1-DARPin-F5 complex. a** Cartoon representation of the complex structure (PDB code 8HKH). LC/A1 is shown in cyan and DARPin-F5 in orange. The zinc ion is shown as a gray sphere. Superimposition of the LC/A1-DARPin-F5 complex structure to **b** the LC/A1-SNAP25 complex structure (PDB code 1XTG) and **c** the structure of the BoNT/A1 holotoxin (PDB code 3BTA)

reveals that DARPin-F5 is interfering with substrate and belt binding, respectively. α- and β-exosites are encircled. The SNAP25 peptide shown in blue. The belt region of BoNT/A1 is shown in red. N- and C-termini of all proteins are indicated in the colors used for their representation.

catalytic site around the enzyme[47]. Loops 50, 170, 250, and 370 form the boundaries of this large cleft on the enzyme surface that in the neurotoxin holostructure is occupied by the belt. Superimposition of the LC/A1-DARPin-F5 complex structure with the structure of LC/A1 bound to SNAP25 (PDB code 1XTG)[47] shows that DARPin-F5 binds to the region of loop 170, thereby preventing the interaction of SNAP25 with this site (Fig. 2b). Superimposition of the structures of the LC/A1-SNAP25 complex and the full-length BoNT/A1 revealed a striking structural similarity between the substrate and the belt[45]. Accordingly, DARPin could also interfere with the binding of the belt as shown in the superimposition of the LC/A1-DARPin-F5 complex structure with the structure of BoNT/A1 holotoxin (PBD 3BTA) (Fig. 2c).

The most prominent feature seen in the complex structure is a network of interactions involving Glu-171 of LC/A1 and three residues from different loops of the DARPin (Fig. 3a, Supplementary Table S2). These interactions include a salt bridge that is formed between Glu-171 of LC/A1 and Lys-91 of the DARPin, respectively. In addition, Arg-25,

and the phenolic hydroxy group of Tyr-50 of DARPin-F5 form hydrogen bonds with the backbone oxygen and the carbonyl group of Glu-171, respectively. Notably, Arg-25 is conserved amongst DARPins. Based on structural considerations, Glu-171 of LC/A1 seems to be a key residue for the interaction with DARPin-F5, although it is not directly involved in the interaction with the SNAP25 substrate. The importance of Glu-171 of LC/A1 for the interaction with DARPin-F5 was demonstrated by mutagenesis and resulted in a > 7000-fold higher $K_D$ value (see next section).

Moreover, two additional residues from LC/A1, Lys-128 and Asp-131, are also involved in the complex formation with DARPin-F5 (Supplementary Table S2). Like Glu-171, these residues are not interacting with the substrate nor the belt, indicating that the binding of the substrate and possibly also belt is prevented through steric hindrance by the DARPin.

In addition, the 8xHis tag present in the DARPin (chains C and D) also interacts with the LC/A1 (chains B and A, respectively) of the other

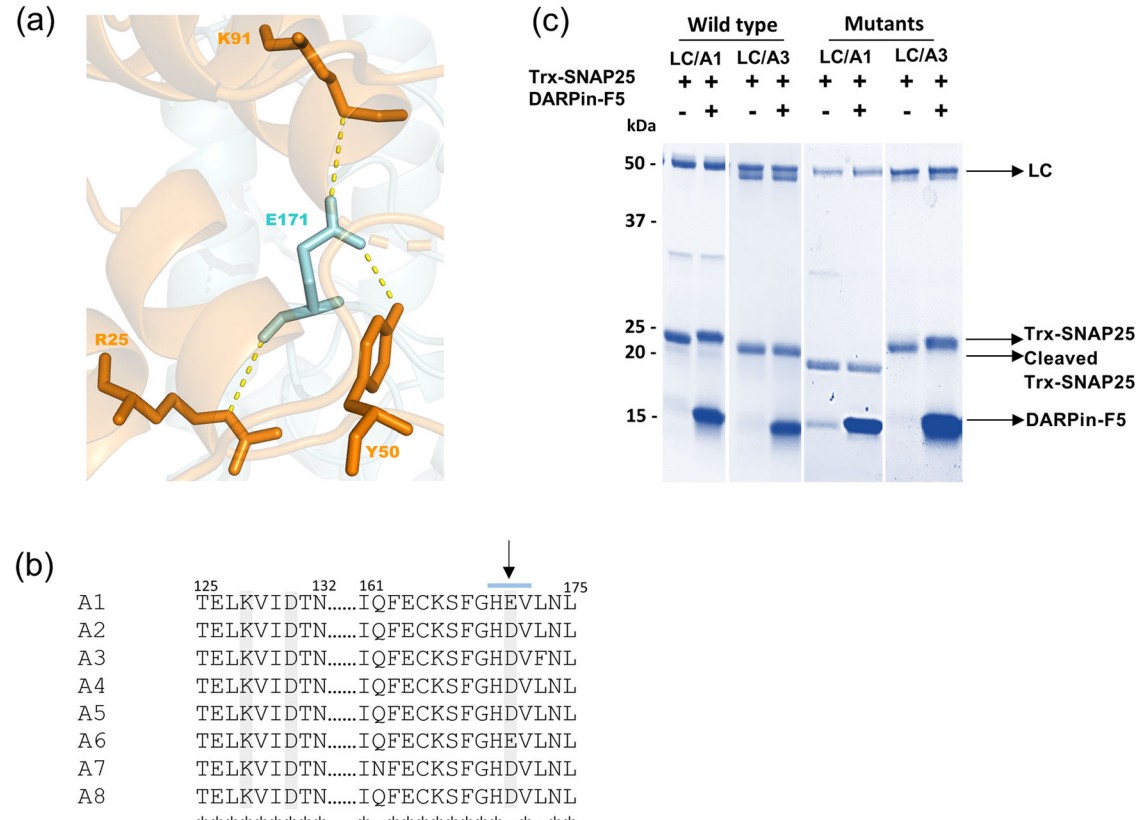

**Fig. 3 | Subtype-specific inhibition of BoNT/A by DARPin-F5. a** Representation of the most prominent interaction between Glu-171 of LC/A1 (cyan) and Lys-91, Arg-25, and Tyr-50 of DARPin-F5 (orange). **b** Conservation of the DARPin-F5 binding regions of LC/A1 (Thr-125 to Asn-132 and Ile-161 to Leu-175) and the corresponding regions of the other seven BoNT/A subtypes. Glu-171 (indicated by an arrow) is present only in BoNT/A1 and A6 while in the other subtypes an aspartate is found at the corresponding position. The other two interacting residues, LC/A1 Lys-128 and Asp-131, are identical in all subtypes. Clustal Omega was used for generating the sequence alignment. *, identical residue; :, strong similarity. Loop 170 is indicated by the cyan line. **c** The impact of DARPin-F5 on the catalysis of wild-type LC/A1 and LC/A3 (residues Pro-2 to Lys-444) and their respective mutants (LC/A1-Glu171Asp and LC/A3-Asp171Glu). DARPin-F5 was not able to inhibit substrate cleavage by LC/ A3 (gel 2, lane 2) and LC/A1-Glu171Asp (gel 3, lane 2). In contrast, DARPin-F5 completely blocked the protease activity of LC/A1 (gel 1, lane 2) and LC/A3-Asp173Glu (gel 4, lane 2). Recombinant wild-type and mutant LCs, cleaved, uncleaved substrate and DARPin-F5 are indicated by arrows. Experiments were repeated twice independently and yielded similar results.

complex (Supplementary Fig. S2). Residues His-11 to Gly-13 form a short β-strand that interacts in an antiparallel manner with the β-strand formed by residues Gly-255 to Phe-260. As a result of this interaction, the N-terminus (His-7 to His-10) points deep into the active site and extends towards the catalytic zinc. This second interaction is unlikely to contribute to the inhibition of LC/A1, because there is no evidence of tetrameric complexes in solution (Supplementary Fig. S3). The interaction is therefore probably needed to establish the crystal packing. This conclusion is consistent with isothermal titration calorimetry (ITC) measurements of DARPin-F5 with and without the 8xHis tag, which revealed very similar binding (Supplementary Fig. S4). Together with the mutagenesis data (see section below), these results also strongly suggest that the 8xHis tag has no influence on the LC/A1-DARPin-F5 complex structure.

## DARPin-F5 is a subtype-specific inhibitor of BoNT/A substrate cleavage

As shown in the alignment of Fig. 3b, the DARPin-binding region is well conserved amongst BoNT/A subtypes. Glu-171 of LC/A1 is a conserved Asp in all other subtypes except BoNT/A6. The other two interacting residues, LC/A1 Lys-128 and Asp-131, are identical in all subtypes. The conservation of the DARPin-binding region suggests that DARPin-F5 might inhibit the catalytic activity of all eight BoNT/A subtypes. To test this hypothesis, we assessed whether DARPin-F5 is able to block the enzymatic activity of LC/A3, a potentially very attractive subtype for

medical applications where a shorter onset and duration of action are required[13,48]. We found that DARPin-F5 was not able to inhibit SNAP25 catalysis by LC/A3. Therefore, we next focused on the role of Glu-171 in the inhibition of SNAP25 catalysis and generated LC/A1 and LC/A3 mutants, in which Glu-171 and Asp-171 were exchanged between subtypes. First, we tested the activity of the mutants. LC/A1-Glu171Asp and LC/A3-Asp171Glu cleaved SNAP25 with the same efficiency as the respective wild-type proteins. DARPin-F5 completely blocked SNAP25 cleavage by LC/A3-Asp171Glu (DARPin-F5/toxin molar ratio of 5:1 or 10:1) but did not inhibit substrate cleavage of LC/A1-Glu171Asp (Fig. 3c, Supplementary Fig. S8), demonstrating the importance of this amino-acid residue for DARPin-F5-mediated inhibition of proteolysis.

Taken together, our findings demonstrate that we can generate BoNT/A subtype-specific DARPins that inhibit substrate cleavage. Such DARPins might be of significant interest as diagnostic tools to identify BoNT subtypes in clinical samples.

## DARPin-F5 binds with high affinity to LC/A1

To assess the binding affinity and the kinetic parameters of the interaction of DARPin-F5 with LC/A1 and LC/A3, we performed surface plasmon resonance (SPR) experiments. To this aim, biotin-labeled LC/ A1 and LC/A3 were immobilized on streptavidin chips and different concentrations of DARPin-F5 were used as analyte. For the analysis of the interaction of LC/A1 and LC/A3 with DARPin-F5 (Fig. 4, Supplementary Fig. S5), a 1:1 Langmuir-binding model was applied. Because of

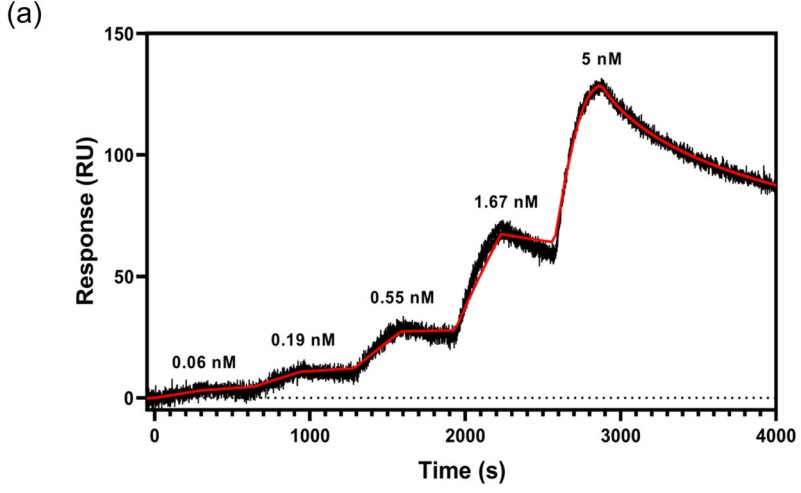

(a)

(b)

| LC/A construct | kon | koff | $K_D$ |
|---|---|---|---|
| | (M-1s-1) | (s-1) | (M) |
| LC/A1 | $5.64 \times 10^6$ | $1.32 \times 10^{-3}$ | $2.38 \times 10^{-10}$ |
| LC/A3 | $1.91 \times 10^4$ | $3.20 \times 10^{-2}$ | $1.68 \times 10^{-6}$ |

**Fig. 4 | Binding affinity and kinetic parameters of the interaction of DARPin-F5 with LC/A1 and LC/A3. a** SPR sensorgram of DARPin-F5 binding to immobilized LC/A1 (black) overlaid with a fit of a 1:1 binding model (red line) using the BiaEvaluation software (Version 4.1). Increasing DARPin-F5 concentrations from 0.06 to 5 nM were applied. **b** Table summarizing the $k_{on}$, $k_{off}$, and $K_D$ values of DARPin-F5 binding to LC/A1 and LC/A3.

the slow dissociation rate, kinetic titration was used for the analysis of the binding of DARPin-F5 to LC/A1. We obtained a $K_D$ value of $2.38 \times 10^{-10}$ M for the binding of DARPin-F5 to LC/A1 (Fig. 4a). In contrast, the interaction of LC/A3 with DARPin-F5 was significantly weaker with a $K_D$ of $1.68 \times 10^{-6}$ M (Fig. 4b, Supplementary Fig. S5), which is consistent with the inability of DARPin-F5 to block substrate cleavage of this subtype. The corresponding association rate constants $k_{on}$ were $5.64 \times 10^6$ and $1.91 \times 10^4$ M$^{-1}$s$^{-1}$ and dissociation rate constants $k_{off}$ of $1.32 \times 10^{-3}$ and $3.20 \times 10^{-2}$ s$^{-1}$ were obtained for the interaction between DARPin-F5 and LC/A1 and LC/A3, respectively. The slow $k_{off}$ value observed for the interaction between LC/A1 and DARPin-F5 explains the strong binding affinity of the inhibitor to the catalytic domain of the subtype. High-affinity binding was also demonstrated by ITC measurements, which revealed that LC/A1 also binds with a similar affinity to DARPin-F5 under acidic conditions (Supplementary Fig. S4).

### DARPin-F5 increases the rate of SNAP25 cleavage by BoNT/A1 in neuronal cells and muscle tissue

We next tested the inhibition potential of DARPin-F5 on BoNT/A1 activity in cells and muscle tissue. Mouse cerebellar granule neurons (CGNs) were incubated with either BoNT/A1 alone or BoNT/A1 pre-incubated with DARPin-F5. Increasing concentrations of BoNT/A1 were used while the DARPin-F5 concentration was kept constant. Cells were lysed and the SNAP25 content was estimated with an antibody that recognizes both the full-length and the BoNT/A1-cleaved forms of SNAP25. Unexpectedly, we found that DARPin-F5 increased the rate of substrate cleavage by BoNT/A1 in neurons (Fig. 5a). At the lowest toxin concentration, virtually no SNAP25 cleavage occurred while in the presence of DARPin-F5, approximately 50% of SNAP25 was cleaved.

We further confirmed the increase in the rate of substrate cleavage of BoNT/A1 mediated by DARPin-F5 using the mouse phrenic nerve (MPN) hemidiaphragm assay, an ex vivo muscle paralysis model that represents the standard method to assay the neuroparalytic activity of BoNTs at the neuromuscular junction. In this experimental setup, BoNTs induce a decrease in the twitch capability of the diaphragmatic muscle by exerting their metalloprotease activity on the phrenic nerve. This decay is followed over time to evaluate BoNT potency but is also used to determine the inhibitory activity of antitoxins.

The effect DARPin-F5 on the paralytic action of BoNT/A1 at the hemidiaphragm was assessed by treating muscles with either 10 pM BoNT/A1 or the DARPin/toxin complex (molar ratio of 10:1 or 25:1). As shown in Fig. 5b, 250 pM DARPin-F5 was unable to prevent the paralytic action of BoNT/A1. However, as observed in cells, DARPin-F5 accelerated toxin action and reduced the time of the halfparalysis by 60%. In contrast, the DARPin alone had no effect on the muscle under the experimental conditions tested.

### DARPin-F5 increases BoNT/A1 translocation leading to a faster onset of toxin action

To understand the molecular mechanism by which DARPin-F5 increases the rate of SNAP25 cleavage of BoNT/A1 in neurons and in the MPN hemidiaphragm assay, we first tested whether DARPin binds to the unreduced full-length toxin. We were not able to obtain data on the binding of DARPin-F5 to full-length toxin by several methods, which is most likely mainly the result of a combination of DARPin-F5 binding and belt dislocation, which might also cause conformational changes in the toxin. Our ITC data look like a mixture of endothermic high-affinity and exothermic low-affinity titrations. However, the quality of the data was never good enough for proper fitting. For MST (Microscale Thermoforesis), we obtained binding data that were impossible to interpret, because the bound/unbound differences change depending on the incubation time. Furthermore, we never reached a plateau of the binding curve at high concentrations, which is necessary for $K_D$ evaluation. For SPR, we observed unspecific binding of DARPin-F5 to the chip matrix that masked a potential signal. As shown in Supplementary Fig. S3, a small shift of the DARPin/BoNT/A1 complex towards a higher molecular weight was observed by size-exclusion

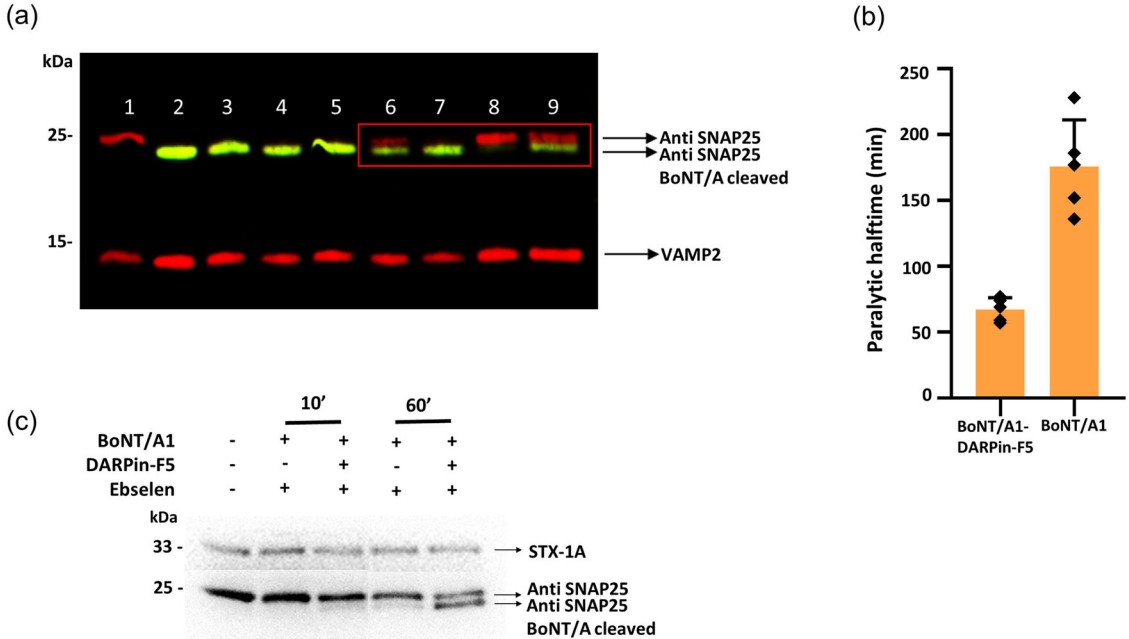

**Fig. 5 | Earlier onset of DARPin-mediated BoNT/A1 action in cells and muscle tissue. a** CGNs were incubated with BoNT/A1 alone (lanes 2, 4, 6, 8) of the BoNT/A1-DARPin-F5 complex (lanes 3, 5, 7, 9) at decreasing BoNT/A1 concentrations of 1, 0.1, 0.01 and 0.001 nM and a constant DARPin-F5 concentration of 3 nM. Lane 1, CGN control without toxin. Cells were lysed and the SNARE content was estimated using the indicated antibodies: SMI-81 (anti-SNAP25) recognizes both the full-length (red) and the cleaved form of SNAP25 (green) and BoNT/A-cleaved recognizes only BoNT/A1-truncated SNAP25. An antibody against intact form of VAMP2 was used as loading control. SNARE proteins recognized by the antibodies are indicated by arrows. The position of marker proteins is indicated. Experiments were repeated three times independently and yielded similar results. **b** The neurotoxicity of 10 pM of BoNT/A1 in the presence of 250 pM DARPin-F5 was assessed in the MPN hemi-diaphragm assay. Data are represented as mean values ± SD, *n* = 5. **c** DARPin-F5 increases the translocation of BoNT/A1. A clear difference in substrate cleavage indicating different rates of translocation was observed in Ebselen-treated cells with or without the DARPin already after 10 min. Cells were lysed and the SNAP25 content estimated as in (**a**). Syntaxin-1A (STX-1A) was used as loading control. The position of marker proteins is indicated. Experiments were repeated three times independently and yielded similar results.

chromatography when compared to the toxin alone. SDS-PAGE analysis of the peak clearly demonstrated the presence of both toxin and DARPin. This result indicates that DARPin-F5 can bind to un-reduced, full-length BoNT/A1 despite the presence of the disulfide bond-anchored belt. We observed that the BoNT/A1-DARPin-F5 interaction still occurs when the complex was transferred to acidic pH (Supplementary Fig. S3), indicating that DARPin-F5 binds to the toxin inside the synaptic vesicles. Together with our findings on an increased rate of SNAP25 cleavage in cells and neuromuscular tissue preparations, it is therefore tempting to speculate that DARPin-F5 enhances LC translocation probably by dislocating the belt, which might result in destabilization of the catalytic domain. Consistent with this hypothesis, nanoscale differential scanning fluorometry (nanoDSF) measurements of BoNT/A1 under neutral and acidic conditions showed a destabilization of the toxin in the presence of DARPin-F5 (Supplementary Fig. S6). This is in agreement with previous publications reporting that stabilization of the BoNT-LC by antibodies or nanobodies can inhibit translocation[34,36,49].

We also performed nanoDSF experiments with LC/A1 with and without DARPin-F5 at physiological and acidic pH (Supplementary Fig. S6). Notably, the unfolding transition of LC/A1 alone occurs over a much broader temperature range compared to the one of the full-length toxin. This finding might reflect the presence of an ensemble of LC/A1 structures that was proposed to exist under physiological conditions[50]. It is also consistent with the presence of a molten globule-like structure of the catalytic domain at acidic pH, which is not observed in the context of the full-length toxin[51]. DARPin-F5 destabilizes LC/A1 at physiological pH but stabilizes the catalytic domain at acidic pH. These results strongly suggest that destabilization of BoNT/A1 is primarily caused by the dislocation of the belt by DARPin-F5.

Reduction of the interchain disulfide bridge mediated by the thioredoxin-thioredoxin reductase system, is a prerequisite for the release of the LC into the cytosol and the subsequent cleavage of SNARE proteins. This process can be blocked by inhibitors of thioredoxin or of its reductase such as Ebselen[52]. To compare the rate of LC/A1 translocation on CGNs, we added Ebselen at various time points after the addition of BoNT/A1 preincubated for at least 1 h with or without DARPin-F5. As shown in Fig. 5c, Ebselen was less effective in blocking SNAP25 cleavage when CGNs were treated with BoNT/A1 preincubated with DARPin-F5. The difference in SNAP25 cleavage was already observable after 10 min and was even more pronounced after 60 min of BoNT/A1-DARPin-F5 incubation. These results clearly indicate that DARPin-F5 increased LC/A1 entry into the neuronal cytosol, which is consistent with a faster and/or more efficient translocation. Furthermore, the findings also indicate that at some step of translocation process, most likely during unfolding of the LC, DARPin-F5 dissociates from the catalytic domain, which is then free to cleave SNAP25.

## Discussion

DARPins are excellent tools for investigating protein structure and function[39]. In the present study, we generated DARPins selected against the catalytic domain of BoNT/A1 and characterized them by biochemical, biophysical, and structural methods together with functional assays in cells and muscle tissue. We identified a DARPin that inhibits BoNT/A1 catalysis in vitro but, unexpectedly, led to an earlier onset of toxin action in cells and muscle tissue.

The existence of multiple subtypes within certain BoNT serotypes is a great public health concern because they are all likely to cause botulism. Accurate identification of BoNTs in clinical samples is therefore a fundamental public health goal. A common strategy to

achieve this aim is to detect the presence of a particular BoNT through its enzymatic activity on a peptide substrate[53]. However, a substantial disadvantage of this approach is that many subtypes are not detected by such peptide substrates, indicating differences in their interactions with the substrate. Towards understanding the characteristic features of substrate cleavage of other serotypes and subtypes at the molecular level, structural knowledge of additional toxin-substrate complexes beyond the two examples, LC/A1-SNAP25[47] and LC/F1-VAMP2[54] that are currently available, seems important.

Our data demonstrate that we can generate subtype-specific LC/A DARPins (Fig. 3). Such subtype-specific DARPins may offer a diagnostic means complementary to activity assays using peptide substrates to detect BoNT-subtype activity in clinical samples.

Surprisingly, DARPin-F5 inhibited SNAP25 cleavage in vitro (Fig. 1) but led to an earlier onset of LC/A1 action in neurons and muscle tissue (Fig. 5). Our current explanation for the earlier onset of BoNT/A1 action is that the DARPin prevents the binding of the belt to the LC in a manner similar to the SNAP25 substrate. Crystal structures of BoNT/A, B, and E demonstrated that the belt wraps around LC in a way almost identical to SNAP25 and VAMP2[20–22]. This hypothesis is supported by the observation that DARPin-F5 interacts with full-length BoNT/A1 at neutral and acidic pH (Supplementary Fig. S3), which would, on the basis of steric hindrance considerations (Fig. 2c), not be possible without a conformational change of the belt region. The loss of the interaction of the belt with LC/A1 results in a destabilization of the toxin (Supplementary Fig. S6), which would then result in faster translocation and subsequent substrate cleavage.

A possible role of the belt in translocation has been recently proposed[29]. It may modulate the conformational change of $H_N$ from a soluble protein to a membrane pore by shielding the hydrophobic BoNT-switch. Synaptic vesicle acidification is likely to result in unfolding of the belt, a process that may help to release the BoNT-switch and initiate the first step of membrane integration of $H_N$. Dislocation of the belt by DARPin-F5 may therefore also have an impact on $H_N$ membrane pore formation and therefore also translocation. This hypothesis is also consistent with a recent report from Lam and colleagues[34] who studied single-chain camelid antibodies that block the translocation of BoNT/E1. They observed a high flexibility of the belt and concluded that this flexibility might contribute to the fast translocation of BoNT/E1 by lowering the energy required for protein unfolding and LC delivery into the cytosol. These conclusions are in agreement with our findings that show destabilization of BoNT/A1 in the presence of DARPin-F5 (Supplementary Fig. S6). Unfolding of LC/A1 will result in a loss of the interaction with DARPin-F5, which will remain in the lumen of the synaptic vesicle.

DARPin-mediated destabilization of the toxin is consistent with our BoNT/A1 chase experiment with Ebselen, which revealed faster translocation of the toxin in the presence of DARPin-F5 (Fig. 5c). It is also in agreement with a publication proposing that successful translocation requires LC/A1 destabilization and molten-globule formation at acidic pH[55]. Consistent with this proposal, it has been recently demonstrated that a nanobodies specific for LC/E1[34] or LC/A1 3[36] or a Fab that bind with high affinity to LC/A1[49] efficiently inhibited translocation of the toxin's catalytic domain, indicating that the stability of LC is critical for translocation. Furthermore, it has been reported that fusion of GFP, which has a stability of ~70 °C at acidic pH, to the N-terminus of BoNT/D resulted in translocation with very low efficiency when compared to other, less stable cargo proteins[56]. However, work remains to be done to clarify whether thermal or mechanical protein stability is the force relevant for LC translocation.

Another factor that has been implicated in faster translocation is the conformation of the full-length toxin. The crystal structures of BoNT/A1 and BoNT/B revealed an open linear arrangement of the three domains with no contact between LC and $H_C$[20,22]. The two domains are separated by $H_N$ that wraps around LC as the belt and then folds into the elongated helical translocation domain. In contrast, the crystal structure of BoNT/E shows a more compact arrangement of domains[21]. LC and $H_C$ are positioned on the same side of $H_N$ with interactions between all domains. An identical organization of domains is also observed in the cryo-EM structures of BoNT/B and BoNT/E[23]. It has been suggested that the unique domain arrangement of BoNT/E is the likely reason for its faster translocation and onset of action[21]. It has also been speculated that for successful translocation the open form of BoNT/A and B will need to convert into closed translocation-competent conformation seen in BoNT/E. This hypothesis was confirmed in recent cryo-EM studies on TeNT, in which the open conformation could be switched into a more compact form upon acidification[57]. The faster translocation of LC/A1 in the presence of the DARPin might therefore possibly be explained by a faster conformational change of BoNT/A1 from the open into the closed compact form upon DARPin binding.

Currently, our knowledge on the role of toxin conformation in translocation is limited, which highlights the need for additional studies to investigate the impact of toxin conformation on translocation.

BoNT/A1 is successfully used as a therapeutic protein for the treatment of a steadily increasing number of neurological and non-neurological disorders such as strabism, chronic migraines or spasticity, as well as cosmetic applications. To date all cosmetic and clinical applications are limited to the use of BoNT/A1 and to a minor extent BoNT/B1[6,7,58]. Serotypes BoNT/E and BoNT/F are, however, potentially very attractive serotypes for applications in therapeutic areas where a shorter onset and duration of action than those of BoNT/A1 are required[58]. For example, BoNT/E has a fast onset of action of about 24 h compared to 3–7 days for BoNT/A1, which would be of tremendous benefit for particular applications[59]. Our findings demonstrate that in the presence of DARPin-F5 BoNT/A1 has a significantly shorter onset of action and therefore offer the possibility to develop a faster acting toxin based on DARPin technology that still has a persistence of 3-4 months characteristic of BoNT/A1.

## Methods

### Ethical statement

All procedures were performed in accordance with the Italian laws and policies (D.L. no 26 14th March 2014), with the guidelines established by the European Community Council Directive no 2010/63/UE and approved by the veterinary services of the University of Padova (O.P.B.A.-Organismo Preposto al Benessere degli Animali) (protocol 359/2015). All the procedures were utilized according to the ethical standards of the Institution where experiments are carried out.

### Ribosome display selection of DARPins binding LC/A1

To generate DARPin binders, biotinylated LC/A1 was immobilized on either MyOne T1 streptavidin-coated beads (Pierce) or Sera-Mag neutravidin-coated beads (GE), depending on the particular selection round, alternating them to avoid binders against either beads. Ribosome display selections were performed essentially as described[41], using a semi-automatic KingFisher Flex MTP96 well platform. To enrich for binders with high affinities, selections were performed over four rounds with decreasing target concentration (round 1:250 pmol target (or 500 nM)); round 2:125 pmol target (or 250 nM); round 3:5 pmol biotinylated target (or 10 nM) with 300 fold excess of non-biotinylated competitor as an off-arate selection; round 4:50 pmol target (or 100 nM) and increasing washing steps. Successively enriched pools were cloned as intermediates in a ribosome display-specific vector.

The library includes N3C-DARPins with the original randomization strategy as reported[60] but includes a stabilized C-cap[39,61,62]. Additionally, the library is a mixture of DARPins with randomized and

non-randomized N- and C- terminal caps, respectively[39,63] and successively enriched pools were cloned as intermediates in a ribosome display-specific vector. Selections were performed over four rounds with decreasing target concentration and increasing washing steps and the third round included a competition with non-biotinylated LC/A1, to enrich for binders with high affinities.

## Screening of DARPins binding LC/A1

The final enriched pool of cDNA coding for putative DARPin binders was cloned into a bacterial pQE30 derivative vector (Qiagen), containing a T5 lac promoter and lacIq for expression control, as fusion construct with an N-terminal MRGS(H)$_6$ tag and C-terminal FLAG tag via unique BamHI and HindIII sites. After transformation of *E. coli* XL1-blue, 380 single DARPin clones selected to bind LC/A1 were expressed in 96-well format by addition of 1 mM IPTG and lysed by addition of B-Per Direct detergent plus Lysozyme and Nuclease (Pierce). After centrifugation these crude extracts were used for initial screening to bind LC/A1 using HTRF. Binding of the FLAG-tagged DARPins to biotinylated LC/A1, was measured using FRET (donor: Streptavidin-Tb cryptate (610SATLB, Cisbio), acceptor: mAb anti FLAG M2-d2 (61FG2DLB, Cisbio)). Further HTRF measurement against 'No Target' allowed for discrimination of LC/A1-specific hits. Experiments were performed at room temperature in white 384-well Optiplate plates (PerkinElmer) using the Taglite assay buffer (Cisbio) at a final volume of 20 µl per well. FRET signals were recorded after an incubation time of 30 min using a Varioskan LUX Multimode Microplate (Thermo Scientific). HTRF ratios were obtained by dividing the acceptor signal (665 nm) by the donor signal (620 nm) and multiplying this value by 10,000 to derive the 665/620 ratio. The background signal was determined by using reagents in the absence of DARPins.

From the initial hits, 32 were chosen and DNA sequence determined by Sanger sequencing. 25 DARPins binding to LC/A1 were identified as single clones. These were expressed and IMAC purified for hit validation ELISA and SEC analysis.

## Purification of DARPins and hit validation

For IMAC purification of the identified 25 DARPins binding to LC/A1 were expressed in small-scale deep-well 96-well plates, lysed with Cell-Lytic B (Sigma) and purified over a 96-well IMAC column (HisPur™ Cobalt plates, Thermo Scientific) including washing with high salt (1 M NaCl) and low salt (20 mM NaCl) PBS buffer. Elution was performed with PBS, 400 mM NaCl, 250 mM imidazole.

ELISA was performed with IMAC-purified DARPins against the biotinylated target protein at a concentration of 50 nM on neutravidin-coated 384 wells, or neutravidin only as control. Detection of DARPins at a concentration of 50 nM was performed using a mouse-anti-FLAG M2 monoclonal antibody (dilution 1:5000; Sigma, F1804) as primary and a goat-anti-mouse antibody conjugated to an alkaline phosphatase (dilution 1:10,000; Sigma, A3562) as secondary antibody. After addition of pNPP (para-nitrophenyl phosphate), absorbance at 405 nm was determined after 30 min. Signals at A540 nm were subtracted as background correction.

## Construct design and cloning

Codon-optimized synthetic DNA fragments encoding active LC/A1 (UniProtKB entry P0DPI1) variants spanning residues Pro-2 to Gly-421, Pro-2 to Gly-433, and Pro-2 to Lys-448 and active LC/A3 variants (UniProtKB entry D3IV24) comprising residues Pro-2 to Gly-417 and Pro-2 to Lys-444 were cloned into the BamHI/EcoRI site of versions of the expression vectors pET-15b or pET-20b for bacterial production. The pET-15b vector was modified in house to contain a N-terminal MKKHHHHHHGSLVPRGS tag and a different multiple cloning site, and in pET-20b, the pelB leader sequence was replaced by a N-terminal MAHHHHHHGS tag. A codon-optimized synthetic gene fragment

encoding residues Met-146 to Gly-204 of human SNAP25 (UniProtKB entry P60880) was cloned into the BamHI/EcoRI site of pHisTrx2, a pET-based expression vector containing an N-terminal 6xHis-tagged thioredoxin A (TrxA) fusion protein[64].

For biotin-labeled full-length active LC/A1 and LC/A3, an N-terminal Avi-tag (GLNDIFEAQKIEWHE) was introduced via PCR and subsequently the DNA fragments were cloned into the BamHI/XhoI site of the pRSFDuet-1 vector. The active mutants LC/A1-Glu171Asp and LC/A3-Asp171Glu were generated from plasmids encoding the full-length wild-type domains using a modified Quikchange method based on the protocol of Zheng et al.[65]. A DARPin-F5 variant with a 8xHis tag that can be removed by thrombin cleavage and without the Flag tag was generated by PCR (Supplementary Fig.S1). The insert was cloned into the BamHI/HindIII site of the original vector. All DNA constructs were sequence-verified (Eurofins). All recombinant proteins mentioned in the text contain an N-terminal 6xHis tag unless otherwise stated.

The oligonucleotides used for construct design are shown in Supplementary Table S3.

## Protein expression and purification

Protein expression and purification were performed as described previously[66]. Proteins were expressed in *E. coli* strain BL21(DE3) (NEB). Bacteria were cultured at 37 °C in LB medium containing appropriate antibiotics for selection until an OD$_{600}$ of 0.6 was reached. The temperature was then lowered to 18 °C, expression was induced with 1 mM IPTG, and incubation continued at 18 °C for -16 h. The cells were harvested by centrifugation (4000 × *g*, 4 °C, 15 min) and stored at −80 °C until further use.

The His-tagged proteins were purified by Ni-NTA affinity chromatography (GE Healthcare) using a buffer containing 50 mM Tris, pH 7.5, 400 mM NaCl, and 20 mM imidazole. After the washing step with 10 CV (column volumes), the proteins were eluted with a high-imidazole buffer (50 mM Tris, pH 7.5, 400 mM NaCl, and 400 mM imidazole). Pooled fractions of eluted protein were subjected to size-exclusion chromatography on a Superdex 200 column (GE Healthcare) in a buffer containing 20 mM HEPES, pH 7.5, and 150 mM NaCl. For complex formation, LCs and DARPins were combined in a 1:1 ratio, co-purified by size-exclusion chromatography (Superdex 200, GE Healthcare) and the pooled fractions concentrated to 8–15 mg/ml for crystallization experiments. The 8xHis tag of the DARPin variant was removed by thrombin cleavage as described previously[64].

The Avi-tagged catalytic domains were expressed according to the protocol from Avidity Avitag™ Technology. Briefly, Avi-tagged LC/A1 and LC/A3 were co-expressed in *E. coli* strain BL21(DE3) (NEB) with the IPTG-inducible biotin ligase BirA. Bacteria were cultured in TYH media supplemented with 10 µg/ml chloramphenicol, 50 µg/ml kanamycin and 20% glucose. When OD$_{600}$ of 0.6 was reached, a biotin solution to a final concentration of 50 µM final was added, protein expression was induced with 1 mM IPTG, and incubation continued for 3 h at 37 °C. Cells were harvested by centrifugation and the Avi-tagged proteins were purified as described above. Sample purity and identity were assessed by SDS-PAGE (Bio-Rad) analysis and Western Blot (Bio-Rad). Protein concentration was estimated by UV at 280 nm and proteins were aliquoted and flash frozen in liquid nitrogen and stored at −80 °C until further use.

## Protease activity assay

Experiments were carried out in a 20 µl reaction volume in TBS-150 (20 mM Tris-HCl pH 7.4, 100 mM NaCl). 20 µM of recombinant LCs were incubated with the selected DARPins at different molar ratios (DARPin/toxin ratio of 50:1, 25:1 and 5:1) for 1 h on ice. 50 µM of purified Trx-SNAP25 was then added to the mixture and incubation continued for 1 h at 37 °C. The enzymatic reaction was stopped by adding SDS-PAGE loading buffer and heating for 5 min. Samples were subjected to

AnyKD gradient SDS-PAGE (Bio-Rad) and gels were stained with Coomassie Blue. The inhibitory effect of the selected DARPin-F5 on LC/A1 and on full-length BoNT/A1 was tested also at molar ratio 2.5:1 (DARPin/toxin). Prior to incubation with DARPin-F5, 1 µM BoNT/A1 was reduced with 10 mM DTT for 30 min at 37 °C.

## Crystallization and structure determination

Full-length and truncated LC/A1 variants in complex with DARPin-F5 were concentrated to 8–15 mg/ml and set up for crystallization at 20 °C using the sitting-drop vapor diffusion method. Proteins were mixed with the mother liquor in a volume ratio of 1:1 and 2:1. Crystals were only obtained for LC/A1 (residue Pro-2 to Gly-421) in complex with DARPin-F5 in 0.1 M HEPES pH 7.5 containing 28% w/v of jeffamine ED-2003. Crystals typically appeared within 3 days and grew to their maximum size within 1–2 weeks. A dataset was collected from single, cryo-cooled crystals diffracting to a resolution of 2.5 Å at beamline PXIII (Swiss Light Source, Villigen, Switzerland) equipped with an EIGER 16 M high-resolution detector (Dectris). Raw data were processed and scaled with XDS[67]. The structure of the LC/A1-DARPin-F5 complex was solved by molecular replacement using the LC/A1-SNAP25 structure (PDB code 1XTG) as a search model[47]. The structure was built and refined using PHENIX. Manual adjustments of the model were done using COOT[68]. Crystallographic data and statistics are presented in Supplementary Table S1. The figures were generated with PyMOL (Schrödinger, LLC, New York).

## Surface plasmon resonance (SPR)

The binding kinetics of the LC/A1-DARPin-F5 interaction were determined with a kinetic titration approach by surface plasmon resonance spectroscopy (SPR) on a ProteON XPR36 instrument (Bio-Rad)[69]. A NLC sensor chip (Bio-Rad) was used, and all experiments were performed in 10 mM HEPES pH 7.0, 150 mM NaCl, 0.005% Tween 20. Two ligand channels were coated each with 800 RU of in vivo biotinylated LC/A1. Five increasing concentrations of 0.06, 0.19, 0.55, 1.67, and 5 nM of DARPin-F5 were injected consecutively onto the interaction surface in duplicates for 300 s at a flow rate of 80 µL/min, followed by a dissociation phase of 5 min, after which the next higher concentration was injected. The dissociation phase of the highest concentration was set to 2 h. For data analysis, the measured signals were double referenced using the ProteOn manager software (Version 3.1.0.6) and fitted to a kinetic titration model using the BiaEvaluation software (Version 4.1.).

The binding kinetics of the LC/A3-DARPin-F5 interaction were determined using an OpenSPR instrument (Nicoya). A Streptavidin sensor chip (Nicoya) was used, and the binding experiments were performed in the same buffer that was used for LC/A1. Following equilibration, the two ligand channels were coated each with 400-600 RU of in vivo biotinylated LC/A3 (or LC/A1 as a positive control). Four increasing concentrations of 0.2, 1, 3, and 5 µM of DARPin-F5 were injected as described above, followed by a short dissociation phase, after which the next higher concentration was injected. The measured signals were analyzed with the Tracedrawer Software (Ridgeview Instruments AB) to obtain $K_D$, $k_{on}$, and $k_{off}$ using a kinetic evaluation assuming a 1:1 binding interaction.

## Isothermal titration calorimetry (ITC)

Binding of the DARPin-F5 variants to LC/A1 was measured on a Affinity ITC instrument (TA Instruments) at 25 °C and 1000 rpm. 200 µl of LC/A1 (typically concentrated to 25–35 µM in PBS buffer pH 7.4 or pH 5.5) was added to the cell. Binding was measured upon addition of the DARPin in a stepwise manner, typically 25 injections of 1.8 µl each, at a concentration 10–15 times higher the protein concentration in the cell. The first titration was 0.5 µl and was subsequently deleted in the data analysis. The analysis was performed using the software (NanoAnalyze) provided by the manufacturer.

## Nanoscale Differential Scanning Fluorometry (nanoDSF)

Real-time simultaneous monitoring of the ITF (Intrinsic Tryptophan Fluorescence) at 330 nm and 350 nm for BoNT/A1 alone and in complex with DARPin-F5 (in an equimolar ratio) was carried out in a Prometheus Panta instrument (NanoTemper Technologies) with an excitation wavelength of 280 nm. Capillaries were filled with 10 µl of sample (1 mg/ml in PBS pH 7.4 or pH 5.5), placed into the sample holder and the temperature was increased from 25 to 90 °C at a ramping rate of 1 °C/min. The ratio of the recorded emission intensities (Em350 nm/Em330 nm), which represents the change in Trp fluorescence intensity was plotted as a function of the temperature. The fluorescence intensity ratio and its first derivative were calculated with the manufacturer's software (PR.ThermControl, version 1.2). Three independent measurements were carried out for each condition.

## Neuronal cell culture and intoxication assay

Primary cultures of rat cerebellar granule neurons (CGNs) were prepared from 6 to 8 days-old rats as previously described[70]. Briefly, cerebella were isolated, mechanically dissociated and trypsinized in the presence of DNase I. Cells were then collected and plated onto 24-well plates pre-coated with poly-L-lysine (50 µg/mL) at a cell density of $4 \times 10^5$ cells per well. Cultures were maintained at 37 °C, 5% $CO_2$, 95% humidity in BME (Basal Medium Eagle, Life Technologies) supplemented with 10 % fetal bovine serum, 25 mM KCl, 2 mM glutamine and 50 µg/mL gentamicin (hereafter indicated as complete culture medium). To arrest the growth of non-neuronal cells, 10 µM of cytosine arabinoside was added to the complete culture medium 18–24 h after plating. CGNs at 6 days in vitro (DIV) were treated with 0.1, 0.01, 0.005 nM of BoNT/A1, alone or preincubated with 3 nM of DARPin in HBS buffer for 2 h at 4 °C. The incubation was prolonged on for 12 or 24 h at 37 °C. To monitor the rate of LC translocation, CGNs were treated with 10 nM of BoNT/A1, alone or preincubated with 3 µM of DARPin-F5 in HBS buffer for 2 h at 4 °C. Subsequently, the mix was diluted in complete culture medium supplemented with 60 mM KCl and CGNs were treated with the mix for 5 min. After the indicated time points 30 µM of Ebselen was added in fresh medium and the incubation was carried on for 12 h at 37 °C. The specific proteolytic activity against SNAP25 was evaluated using immunoblotting and imaging with specific antibodies.

## Immunoblotting

Cells were directly lysed with Laemmli sample buffer supplemented with protease inhibitors (Roche). Cell lysates were loaded onto NuPage 4–12% Bis-Tris gels (Life Technologies) and separated by electrophoresis in MOPS buffer. Proteins were transferred onto Protran nitrocellulose membranes (Whatman) and saturated for 1 h in PBS-T (PBS, 0.1% Tween 20) supplemented with 5% non-fatty milk. Anti-SNAP25 (SMI-81, 1:10,000) was from Biolegend; anti-VAMP-2 (104211, 1:2000) was from Synaptic System; anti-SNAP25 BoNT/A-cleaved (1:5000) was home made and used as previously described[71]; anti-Syntaxin-1A/1B (1:2000) was home made and used as previously described[72]. Incubation with indicated primary antibodies was performed overnight at 4 °C. The membranes were then washed three times with PBS-T and incubated with appropriate HRP-conjugated secondary antibodies (Anti mouse 1:5000 and Anti rabbit& mouse Flourescent labeled 1:2000) for 1 h. Membranes were washed three times with PBS and proteins revealed with an Uvitec gel doc system (Uvitec Cambridge).

## Mouse phrenic nerve (MPN) hemidiaphragm assay

The experiments were performed in accordance with the Italian laws and policies (D.L. no 26 14th March 2014), with the guidelines established by the European Community Council Directive no 2010/63/UE and approved by the Italian Ministry of Health and veterinary services of the University of Padova (O.P.B.A.-Organismo Preposto al Benessere degli Animali) (protocol 359/2015). Swiss-Webster adult male CD-1

mice (20–25 g, 3 months old) were housed under controlled light/dark conditions, and food and water were provided ad libitum. The isolated mouse diaphragms were prepared from CD-1 mice and halved into two contralateral hemi-diaphragms still innervated with their own phrenic nerve. Muscles were mounted into two chambers filled with 4 ml of oxygenated (95% $O_2$, 5% $CO_2$) solution (139 mM NaCl, 12 mM NaHCO$_3$, 4 mM KCl, 2 mM CaCl$_2$, 1 mM MgCl$_2$, 1 mM KH$_2$PO$_4$ and 11 mM glucose, pH 7.4). The two phrenic nerves were stimulated via two ring platinum electrodes with supramaximal stimuli of 3 V amplitude and 0. 1 ms pulse duration, with a frequency of 0.1 Hz. Muscle contraction was monitored with an isometric transducer (Harvard Apparatus); data were recorded and analyzed via an iWORX 118 system with Labscribe software (Harvard Apparatus). DARPin-toxin mixtures were prepared at three different molar ratios of 10:1, 25:1, and 50:1 and added directly to the oxygenated solution of one muscle, and the same volume of toxin at 10 pM final concentration was added alone to the contralateral one for direct comparison. The twitch was monitored until complete paralysis was achieved. Graphs show muscle twitching capability over time, reported as a percentage with respect to the initial value obtained before toxin addition.

## Reporting summary

Further information on research design is available in the Nature Portfolio Reporting Summary linked to this article.

## Data availability

Atomic coordinates of the LC/A1-DARPin-F5 complex were deposited in the Protein Data Bank database under accession number 8HKH. Structures used for comparative analysis in this manuscript can be found with the following PDB accession codes: 3BTA for the BoNT/A1 holotoxin; 1XTG for the LC/A1-SNAP25 complex structure. All data presented in this manuscript can be found as Source Data in the accompanying Source Data file. Source data are provided with this paper.

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

## Acknowledgements

The PXI beamline staff (Swiss Light Source, Villigen, Switzerland) is acknowledged for their support during the diffraction experiments. We thank Dr. Paola Caccin (University of Padova, Italy) for technical assistance in conducting the MPN hemidiaphragm assays and analysis. Prof. Ornella Rossetto (University of Padova, Italy) is acknowledged for the provision of antibodies. We thank Dr. Birgit Dreier for coordinating the work towards the end of the selection project. This work was supported by grants 31003A_163449 and 31003A_170028 of the Swiss National Science Foundation to R.A.K. and X.L., respectively.

## Author contributions

R.A.K., J.S., A.P. X.L., and M.P. designed the research; O.L., Y.W. S.F., T.R., J. M., and G.Z. carried out the research; O.L., Y.W., M.P, and R.A.K. analyzed the data; O.L. and R.A.K. wrote the manuscript with input from the other authors.

## Competing interests

The authors declare no competing interests.
