## [Peer Review File · Nature Communications]

Reviewers' Comments:

Reviewer #1:

Remarks to the Author:

The paper by Leka, et al describes the engineering and characterization of DARPin F5 for the neutralization of Botulinum neurotoxin/A1. Despite very strong in vitro toxin neutralization activity, DARPin F5 was found unexpectedly to enhance the toxin's toxicity in cell culture. The authors solved the crystal structure of DARPin F5 in complex with the LC/A1 and identified critical residues in both the toxin and DARPin that mediate the interaction. The enhancement of the toxin's activity is unprecedented and was postulated to be due to increase of the translocation of toxin from the acidic endosome to the cytosol. The paper is well written, and the results are significant and important. I only have a few minor comments.

1. Since the epitope of DARPin F5 overlaps with that of the belt, I am wondering whether there is any affinity difference, especially *K_{on}*, between F5 binding to LC/A1 vs. the holotoxin. I presume the *K_{on}* to holotoxin can be slower due to the need to displace the belt region.
2. It is very surprising that, despite the strong in vitro toxin-neutralization activity, F5 enhanced the toxicity of the toxin in cell culture. I think it is a possibility that F5 enhances the toxin translocation, although it is unprecedented and other molecules displacing the belt was found to inhibit translocation. However, to rule out other possibilities, it would be helpful if the authors can determine whether F5 can protect the cells when it is expressed intracellularly.
3. The title "A DARPin increases the Catalytic Activity of Botulinum Neurotoxin A1" is somewhat misleading since all the in vitro assay showed inhibition of the catalytic activity and translocation is not necessarily a catalytic activity. I suggest the authors reconsider the title to better reflect the unique properties of F5.
4. Some of the methods lack sufficient details. For example, the description of 'Ribosome display selection of DARPins binding BoNT/A1/LC' appears to be overly generic. Specifics to this critical step should be included.

Reviewer #2:

Remarks to the Author:

The manuscript describes the application of DARPins (designed ankyrin repeat proteins) to generate serotype-specific inhibition of in vitro BoNT/A1 protease activity. The authors also present an X-ray crystal structure of the DARPin-BoNT/A1 light chain complex which shows binding of DARPin to a site adjacent to the active site and overlapping the SNAP-25 extended substrate binding site. In contrast to these in vitro results, the DARPin increased BoNT/A1 activity in neurons and in a mouse phrenic nerve hemidiaphragm assay. As detailed below the authors model that DARPin-F5 increases translocation of the toxin is only indirectly supported and the mechanism of action cannot be clearly described. Moreover, the presence of the His6 tag from one protomer of BoNT making crystal contacts with the active site of the adjacent protomer, raises concerns about the utility of the crystal structure as a true reflection of that in solution. Overall these concerns about the structure and lack of support for the mechanism of action lower the potential impact of the findings.

Please see the following detailed points (in order of appearance using pages from PDF for review).

Page: 8- "The belt binds to LC/A1 in the full-length toxin in a manner that is very similar to the interaction of LC/A1 with the substrate [45]. " This statement is very cryptic- in the following section the authors do not show a close up view of interactions that are similar or list which residues on the DARPin make similar interactions as substrate- which would support this model. The authors later state "Based on structural considerations, Glu-171 of LC/A1 seems to be a key residue for the interaction with DARPin-F5, although it is not directly involved in the interaction with the SNAP25 substrate." Which seems the opposite of the previous statement.

Page 8- "the 6xHis tag present in the DARPin (chains C and D) also interacts with the LC/A1 (chains B and A, respectively) of the other complex" The interaction extends into the active site and interacts with the catalytic Zinc ion. I agree this is unlikely to occur in solution but it also raises the major concern that the pose of the LC with the DARPin may not be that seen in solution

either as the N-term is constrained in the crystal contact.

Page: 11- The authors state "We observed that the BoNT/A1-DARPin-F5 interaction still occurs when the complex was transferred to acidic pH (Supplementary Figure S2), indicating that DARPin-F5 binds to the toxin inside the synaptic vesicles. Together with our findings on increased toxin activity in cells and neuromuscular tissue preparations, it is therefore tempting to speculate that DARPin18 enhances LC translocation probably by dislocating the belt, which might result in destabilization of the catalytic domain" These are two large leaps given the slim evidence of a small shift in retention on size exclusion chromatography.

Page 11- The results with Ebselen do show enhanced cleavage when BoNT is pretreated with DARPin-F5 but the findings really do not provide reasonable support for proposing any mechanism of action. Overall the in cell findings are extremely phenomenological and the experiments presented do not support the mechanisms proposed. This is especially problematic as the resulting enhancement in activity is the opposite of the inhibitory effects of DARPin-F5 in vitro. The lack of support for the in-cell mechanism proposed, together with the concerns raised by the effect of the N-terminal His tag binding on crystal packing and the observed DARPin-BoNT complex make the report much less compelling.

In table S1 B-factors should be broken into average for the protein, the liganding DARPin and the solvent. Are the B-factors of DARPin similar to those of BoNT LC?

Minor corrections- typos

Page: 3- "Furthermore, there exist mosaic toxins between serotypes C and D [14] and partially between A and F [15]." this sentence seems misplaced- it seems it should be moved before the preceding sentence.

Page: 9- Taken together, our findings demonstrate that we can generate BoNT/A subtype-specific DARPins that inhibit substrate catalysis" This would be better worded "... inhibit substrate cleavage"

Reviewer #3:

Remarks to the Author:

Botulinum neurotoxin A1 (BoNT/A1) is widely used in clinic for treating a variety of neurological disorders. It is also a potential bioterrorism threat agent. Here the authors screened and identified DARPins (Designed Ankyrin Repeat Proteins), which are small (14-18 kDa) protein binders, using ribosome display approach. One DARPin (DARPin-F5) is fully characterized in this study, which blocks the protease activity of BoNT/A1 in vitro. The authors further solved the co-crystal structure of this DARPin-F5 in complex with the protease domain of BoNT/A1 (LC/A1), revealing that DARPin-F5 blocks LC/A1 by binding to the region between exosites critical for recognizing the native substrate. Surprisingly, the authors found that co-incubate of this DARPin-F5 with BoNT/A1 led to apparently higher activity of BoNT/A1, resulting in more cleavage of the substrate in cultured neurons and faster blockage of muscle contraction in an ex vivo model. The authors carried out additional experiments suggesting that co-incubation with DARPin-F5 may facilitated translocation of BoNT/A1 into the cytosol, although this point remains to be fully established. A DARPin binding to the LC/A1 can result in enhanced activity is an eye-opening major finding. It is potentially useful as well. This DARPin will be a valuable tool.

Major:

1. The structure showed interactions with the 6xHis6 tag, can the authors evaluate and compare the binding affinity between tagged and non-tagged DARPin-F5?
2. It will be helpful if the authors can measure and compare the binding affinity for DARPin-F5 to full-length BoNT/A1 versus to LC/A1, as well as at different pH for LC/A1.
3. It will be helpful for authors to include the exact sequence of DARPin-F5 in either figure or supplementary figures, so others can evaluate the work and utilize this tool.

Minor:

Figure 1: add a cartoon to illustrate the screening method and results.

Figure 3c: it will be better to show the uncropped results here.
Abstract needs to be revised, start with a brief introduction to botulinum neurotoxins and to DARPins.
Line 21: "Repeat"
Line 40: extra . before [5].
Line 30: in the United States
Line 109: needs full name for HTRF assay.
Line 221: add (CNGs)
Line 252: DARPIn-F5
Line 258: LC/A1
Line 335: delete this sentence.
Line 338: LC/A1

Reviewer #4:

Remarks to the Author:

In the manuscript entitled "A DARPIn Increases the Catalytic Activity of Botulinum Neurotoxin A1" Designed Ankyrin Repeat Proteins (DARPins) are presented as investigative tools to probe botulinum neurotoxin (BoNT) function and as potential antidotes for botulism. One DARPIn, DARPIn-F5, was found to block SNAP25 cleavage by BoNT/A1 in vitro. X-ray crystallography revealed that DARPIn-F5 inhibited BoNT/A1 activity by interacting with a SNAP25-binding region of the light chain metalloprotease between the α - and β -exosites. However, in contrast to these in vitro results, DARPIn-F5 accelerated BoNT/A1 cleavage of the substrate in primary neurons. This result was confirmed in a mouse phrenic nerve hemidiaphragm assay, which showed faster paralysis in the presence of DARPIn-F5.

The manuscript is well written and the data are convincing. Nonetheless, there are some weaknesses that need to be addressed.

First, the title of the manuscript mischaracterizes the findings. The results show that DARPIn-F5 contributed to faster translocation of the BoNT/A1 light chain to the neuronal cytoplasm, and as a consequence, SNAP-25 cleavage was accelerated resulting in more rapid onset of BoNT-induced neuromuscular action. There is no evidence that DARPIn-F5 contributed to increased enzymatic activity of BoNT/A1 LC metalloprotease. The correct description of the effect of the accelerated LC translocation is very often intertwined with incorrect statements related to "increased activity" (lines 31, 219, 225-227, 228, 243, 251, 272, 287-288, etc). There are "effects" of DARPins all of which are consequences of increased LC translocation. Therefore, all notes and sentences related to the "increased activity" should be revised accordingly.

Second, from the experimental evidence presented in the manuscript it is hard to envision how binding of DARPIn-F5 to LC/A1 in the sub-nanomolar range contributes to accelerated relocation of the metalloprotease through the endosomal pore. This binding should contribute to stabilization of the LC/A1, similarly to the stabilization of the LC/A1 by nanobodies (lines 253-254 of the manuscript) and inability of the metalloprotease in complex with nanobodies to pass through endosomal pore to the neuronal cytoplasm. According to the results shown, the metalloprotease is not only able to translocate with higher efficiency, but also able to cleave the SNAP25 substrate in the neuronal cytosol, thus DARPIn-F5 should remain in the endosome, i.e., should be spatially separated from the enzyme, otherwise LC/A1 would still in complex with DARPIn, and SNAP25 cleavage would be blocked.

Examination of the data presented suggests an explanation for this contradiction, in that the interpretation of the experiment shown in Figure S2 may be a misinterpretation of the data. It is likely that DARPIn-F5 binds the LC/A1 as a single entity and as a part of BoNT/A1 holotoxin at neutral/basic pH (otherwise it is hard to imagine how the DARPIn not only undergoes neuronal uptake with the toxin, but also generates an effect of faster translocation of metalloprotease reported in this manuscript). Thus, perhaps the shift of the protein peak in SEC (Figure 2S) toward higher MW under neutral pH and presence of DARPIn-F5 in the fraction with the holotoxin shown in Figure S2 can be explained through an interaction of DARPIn-F5 with LC/A1. However, the seemingly preserved interaction DARPIn-F5 with LC/A1 at pH 5.2 may have a different explanation. The sequence of DARPIn-F5 was not provided, thus it is not possible to determine the charge of this protein at pH 5.2. The electrostatic charge of BoNT/A1 holotoxin and DARPIn-F5 at pH 5.2 may contribute to a relatively weak ionic interaction between these molecules; in the absence of high

salt in the buffer used for SEC this interaction may be preserved and misinterpreted. It will be necessary to determine if DARPin-F5 still forms a high affinity complex with 1) LC/A1 and 2) with BoNT/A1 holotoxin at an acidic pH. This might be done using an experimental setting similar to that presented in Figure 4 and lines 203-217 of the manuscript.

If the DARPin-F5/LC/A1 interaction is absent/weakened at acidic pH, then observations from other experiments presented in the manuscript can be logically explained: 1) At neutral pH DARPin-F5 can bind both – LC/A1 and BoNT/A1; when it binds to holotoxin, the “belt” enveloping the LC/A1 probably becomes dislocated, as authors mentioned in the manuscript (line 252).

Reference 29 mentioned by the authors (lines 71, 72 of the manuscript) not only discusses “exposure of the hydrophobic peptide through a viral-fusion-peptide-like pH-dependent molecular switch,” but the role of the “belt” as an extended part of this switch, with two important statements: 1) “The belt of BoNT/A1 may modulate the conformational change of HN by interacting with and shielding the hydrophobic surface of the BoNT-switch. The belt is likely unfolded upon membrane binding and vesicle acidification that may help to release the BoNT-switch”; 2) “The HN core lacking the “belt” (termed tHN) was shown to form an ion channel independent of pH”.

The probable dislocation of the “belt” with DARPin-F5 (as authors proposed, line 252) at neutral pH exposes tHN, which, as mentioned above can form an ion channel independent of pH. Endosomal acidification would not only cause dissociation of DARPin-F5 and LC/A1 metalloprotease but would result in “globular melting” of the enzyme allowing it to pass through the narrow endosomal pore already formed before acidification by “uninhibited” tHN. In contrast, if, at an acidic endosomal pH, DARPin-F5 remains associated with LC/A1, then the effect would be similar to the prevention of translocation following stabilization of BoNT-LC by antibodies or nanobodies as mentioned in lines 253-254 and opposite to the observations described in this manuscript.

Minor issues:

Line 21 – Repeat, not Repat.

Line 121 – What is the percent inhibition when the molar ratio is 1:1?

Line 172 – Interesting that all other LC/A subtypes have K128 and D131 preserved. That means a single amino acid – E171 is determinative for the high affinity interaction between DARPin-F5 and LC/A1. Just an absence of a single methylene group in the side-chain (E>D mutation) is able to abolish this binding.

Lines 175-181 – is this explanation really necessary?

Line 258 – to compare, not to estimate.

Lines 259, 261, and elsewhere – how long is the preincubation? Is it necessary to “preincubate” instead of just “mix” the components?

Lines 298, 299 “with a recent publication.” The citation to Cai et al is from 2006. A 17-year-old paper is not recent.

Lines 308-321 – seems not really relevant to the observations reported in the manuscript.

Lines 325 – 335 – faster acting toxin and lower dose of the toxin – are totally different characteristics. By using a faster acting toxin, the onset is faster but the actual dose of the toxin supposed to be the same (unless of course the slow-acting toxin instead of translocating to the cytosol get degraded through late endosome and lysosome), therefore this entire paragraph needs to be rewritten. The formation of an immune response to the toxin is irrelevant because of what was said above. By using DARPin-F5 as a means to provide faster onset may result in formation of immune response to the DARPin-F5 itself (which is used in significant (25 times) molar excess in this manuscript).

Figure S2 – What are the additional protein bands running below un-reduced holotoxin at pH 5.2? Aren't these bands the consequence of unexpected reduction?

POINT-BY-POINT RESPONSE TO REVIEWERS' COMMENTS

Reviewer #1

The paper by Leka, et al describes the engineering and characterization of DARPin F5 for the neutralization of Botulinum neurotoxin/A1. Despite very strong in vitro toxin neutralization activity, DARPin F5 was found unexpectedly to enhance the toxin's toxicity in cell culture. The authors solved the crystal structure of DARPin F5 in complex with the LC/A1 and identified critical residues in both the toxin and DARPin that mediate the interaction. The enhancement of the toxin's activity is unprecedented and was postulated to be due to increase of the translocation of toxin from the acidic endosome to the cytosol. The paper is well written, and the results are significant and important. I only have a few minor comments.

1. Since the epitope of DARPin F5 overlaps with that of the belt, I am wondering whether there is any affinity difference, especially K_{on} , between F5 binding to LC/A1 vs. the holotoxin. I presume the K_{on} to holotoxin can be slower due to the need to displace the belt region.

We tried to determine the binding parameters of DARPin-F5 to the full-length toxin by ITC, MST and SPR but despite numerous attempts, we were not able to obtain conclusive results. We think that these complications are due to presence of the belt that as indicated by the reviewer needs to be displaced before DARPin-F5 can bind to the LC.

However, we generated additional data, which show that DARPin-F5 binds to the toxin under neutral and acidic conditions. Moreover, DARPin-F5 binding results in a destabilization of the toxin as determined by nanoDSF measurements. Destabilization of the toxin is consistent with faster translocation. We included the nanoDSF results in the revised version of the manuscript.

2. It is very surprising that, despite the strong in vitro toxin-neutralization activity, F5 enhanced the toxicity of the toxin in cell culture. I think it is a possibility that F5 enhances the toxin translocation, although it is unprecedented and other molecules displacing the belt was found to inhibit translocation. However, to rule out other possibilities, it would be helpful if the authors can determine whether F5 can protect the cells when it is expressed intracellularly.

CNGs are very hard to transfect. Based on our experience, classical transfection methods using lipofectamine or calcium phosphate would also not result in DARPin yields that would be needed for the biochemical assessment of SNAP-25 cleavage upon BoNT/A1 intoxication. Therefore, one would need to use a virus-based transfection method for DARPin expression, which is typically a rather lengthy process.

Davletov and colleagues transduced LC domains into cells instead of transfecting plasmid DNA (Arsenault et al., J. Neurochem., 2014). To address the comment of the reviewer, we

used this method. However, we didn't observe any protection effect from the transfected DARPin-F5 of neurons treated with BoNT/A. Based on our inhibition results obtained with DARPin-F5, we conclude that transduction of the DARPin into CNGs didn't work.

3. The title "A DARPin increases the Catalytic Activity of Botulinum Neurotoxin A1" is somewhat misleading since all the in vitro assay showed inhibition of the catalytic activity and translocation is not necessarily a catalytic activity. I suggest the authors reconsider the title to better reflect the unique properties of F5.

Reviewer 4 made a similar comment- We agree with both reviewers and changed the title to "A DARPin Promotes Faster Onset of Botulinum Neurotoxin A1 Action".

4. Some of the methods lack sufficient details. For example, the description of 'Ribosome display selection of DARPins binding BoNT/A1/LC' appears to be overly generic. Specifics to this critical step should be included.

The rationale of the ribosome display strategy as carried out here, aimed to obtain DARPins of higher affinity, has been described in reference 41. Briefly, we are lowering the amount of coated target from round to round, as now explicitly stated in Materials and Methods. Thereby, in early rounds, with high amounts of target, many binders of all affinity ranges are being enriched. By lowering the amount of target, in later rounds they are competing for target and equilibration will favor the tighter binders. In the third round (off-rate selection) we are adding the target, after a short delay, in large excess as competitor, such that binders with fast off rate will bind to the non-biotinylated competitor, and only the tightest binders will remain. As this stringent selection drastically decreases the number of molecules, in the last round, they are amplified with more target and no competitor and thus lower stringency.

As this has been discussed in detail in the above reference, which has been cited, we do not believe it appropriate to state this again in this manuscript, as these are not new findings.

Reviewer #2

The manuscript describes the application of DARPins (designed ankyrin repeat proteins) to generate serotype-specific inhibition of in vitro BoNT/A1 protease activity. The authors also present an X-ray crystal structure of the DARPin-BoNT/A1 light chain complex which shows binding of DARPin to a site adjacent to the active site and overlapping the SNAP-25 extended substrate binding site. In contrast to these in vitro results, the DARPin increased BoNT/A1 activity in neurons and in a mouse phrenic nerve hemidiaphragm assay. As detailed below the authors model that DARPin-F5 increases translocation of the toxin is only indirectly supported and the mechanism of action cannot be clearly described. Moreover, the presence of the His6 tag from one protomer of BoNT making crystal contacts with the active site of the adjacent protomer, raises concerns about the utility of the crystal structure as a true reflection of that in solution. Overall these concerns about

the structure and lack of support for the mechanism of action lower the potential impact of the findings.

Please see the following detailed points (in order of appearance using pages from PDF for review).

Page: 8- "The belt binds to LC/A1 in the full-length toxin in a manner that is very similar to the interaction of LC/A1 with the substrate [45]. " This statement is very cryptic- in the following section the authors do not show a close-up view of interactions that are similar or list which residues on the DARPin make similar interactions as substrate- which would support this model. The authors later state "Based on structural considerations, Glu-171 of LC/A1 seems to be a key residue for the interaction with DARPin-F5, although it is not directly involved in the interaction with the SNAP25 substrate." Which seems the opposite of the previous statement.

DARPin-F5 is not binding the same residues of LC/A1 as the substrate or the belt. In the DARPin-binding region, the belt and the substrate bind also to different LC/A1 residues. Accordingly, DARPin-F5 prevents the binding of the substrate and probably also belt by steric hindrance.

We agree with the reviewer that the first statement is confusing. We revised the sentence and now write "Superimposition of the structures of the LC/A1-SNAP25 complex and the full-length BoNT/A1 revealed a striking structural similarity between the substrate and the belt"

Page 8- "the 6xHis tag present in the DARPin (chains C and D) also interacts with the LC/A1 (chains B and A, respectively) of the other complex" The interaction extends into the active site and interacts with the catalytic Zinc ion. I agree this is unlikely to occur in solution but it also raises the major concern that the pose of the LC with the DARPin may not be that seen in solution either as the N-term is constrained in the crystal contact.

We performed additional experiments using a DARPin-F5 variant without the 8xHis tag. We obtained very similar values for the binding of DARPin-F5 with and without the 8xHis tag to LC/A1 using ITC measurements. We included these results in the revised version of the manuscript and Supplementary Figure S4.

The standard method to test whether a complex crystal structure corresponds to the complex in solution is mutagenesis. We reported in the original manuscript that the interactions between Glu-171 of LC/A1 and residues of DARPin-F5 are a prominent feature seen in the complex crystal structure. Mutagenesis of Glu-171 to Asp and SPR binding experiments of the wild-type and mutant catalytic domain demonstrated very clearly that this residue is critical for high-affinity binding and together with the ITC measurements that the complex crystal structure is therefore very likely to correspond to the one in solution. For clarity, we emphasized this point in the revised manuscript.

Page: 11- The authors state “We observed that the BoNT/A1-DARPin-F5 interaction still occurs when the complex was transferred to acidic pH (Supplementary Figure S2), indicating that DARPin-F5 binds to the toxin inside the synaptic vesicles. Together with our findings on increased toxin activity in cells and neuromuscular tissue preparations, it is therefore tempting to speculate that DARPin18 enhances LC translocation probably by dislocating the belt, which might result in destabilization of the catalytic domain” These are two large leaps given the slim evidence of a small shift in retention on size exclusion chromatography.

We agree that the small shift seen in SEC alone is slim evidence and could also be caused by a conformational change of the toxin. The important point is the analysis of the peak by SDS-PAGE, which clearly revealed the presence of both DARPin-F5 and toxin.

To demonstrate an interaction between DARPin-F5 and the full-length toxin by other methods, we performed additional experiments. GST pull-down experiments confirmed our results. We didn't include these results in the revised version of the manuscript because they basically show the same thing as the SDS-PAGE analysis of the SEC peaks. Furthermore, nanoDSF measurements performed with the toxin/DARPin complex demonstrated a destabilization of the toxin in the presence of DARPin-F5 under neutral and acidic conditions (Supplementary Figure S6). Since toxin stability is concentration-independent these results are consistent with the binding of DARPin-F5 to the full-length toxin. We also performed ITC measurements that show binding of DARPin-F5 to LC/A1 under neutral and acidic conditions (Supplementary Figure S4). We included these results in the revised version of the manuscript.

Page 11- The results with Ebselen do show enhanced cleavage when BoNT is pretreated with DARPin-F5 but the findings really do not provide reasonable support for proposing any mechanism of action. Overall the in cell findings are extremely phenomenological and the experiments presented do not support the mechanisms proposed. This is especially problematic as the resulting enhancement in activity is the opposite of the inhibitory effects of DARPin-F5 in vitro. The lack of support for the in-cell mechanism proposed, together with the concerns raised by the effect of the N-terminal His tag binding on crystal packing and the observed DARPIN-BoNT complex make the report much less compelling.-

We agree with the referee that the experiments presented are phenomenological. Yet, we believe they are consistent with an enhanced translocation rate of LC/A favored by DARPin-F5. In fact, inhibitors of the thioredoxin system, like Ebselen, block the cytosolic release of LC/A just after its translocation, when the disulfide bridge linking the light chain and the heavy chain has to be reduced to free the enzyme in the cytosol and to enable its metalloprotease activity against SNAP-25. In the presented experiment, we chased LC/A translocation by adding Ebselen at the indicated time points, thereby obtaining a time-resolved monitoring of LC/A delivery in the cytosol.

We generated additional data that demonstrate binding of DARPin-F5 to the full-length toxin under neutral and acidic conditions. Binding of the DARPin results in destabilization of the toxin, a finding that is generally accepted to be consistent with faster translocation. During translocation, LC/A1 is believed to unfold, which would result in a loss of binding to the DARPin, which would remain in the lumen of the synaptic vesicle. We tried to make this point clearer in the revised version of the manuscript.

There is also the possibility that DARPin-F5 acts on the translocation domain. Please see the second comment of reviewer and the answer to the comment.

In table S1 B-factors should be broken into average for the protein, the liganding DARPin and the solvent. Are the B-factors of DARPin similar to those of BoNT LC?

We updated the table. The B factor (47.75) shown in Figure S1 is the average for macromolecules (47.80), ligands (42.38) and solvent (31.97). The average B factor of 2 DARPin molecules (chain C and D) is 51.71, which is similar to the B factor of the overall structure.

Minor corrections- typos

Page: 3- "Furthermore, there exist mosaic toxins between serotypes C and D [14] and partially between A and F [15]." this sentence seems misplaced- it seems it should be moved before the preceding sentence.

As suggested by the reviewer, we moved the sentence in front of the preceding sentence.

Page: 9- Taken together, our findings demonstrate that we can generate BoNT/A subtype-specific DARPins that inhibit substrate catalysis" This would be better worded ".... inhibit substrate cleavage"

We agree with the reviewer. We replaced "inhibit substrate catalysis" by "inhibit substrate cleavage".

Reviewer #3

Botulinum neurotoxin A1 (BoNT/A1) is widely used in clinic for treating a variety of neurological disorders. It is also a potential bioterrorism threat agent. Here the authors screened and identified DARPins (Designed Ankyrin Repeat Proteins), which are small (14-18 kDa) protein binders, using ribosome display approach. One DARPin (DARPin-F5) is fully characterized in this study, which blocks the protease activity of BoNT/A1 in vitro. The authors further solved the co-crystal structure of this DARPin-F5 in complex with the protease domain of BoNT/A1 (LC/A1), revealing that DARPin-F5 blocks LC/A1 by binding to the region between exosites critical for recognizing the native substrate. Surprisingly, the authors found that co-incubate of this DARPin-F5 with BoNT/A1 led to apparently higher activity of BoNT/A1, resulting in more cleavage of the substrate in cultured neurons

and faster blockage of muscle contraction in an ex vivo model. The authors carried out additional experiments suggesting that co-incubation with DARPin-F5 may facilitated translocation of BoNT/A1 into the cytosol, although this point remains to be fully established.

A DARPin binding to the LC/A1 can result in enhanced activity is an eye-opening major finding. It is potentially useful as well. This DARPin will be a valuable tool. Major:

1. The structure showed interactions with the 6xHis6 tag, can the authors evaluate and compare the binding affinity between tagged and non-tagged DARPin-F5?

Reviewer 2 made a similar comment. Therefore, please also see our answer to his question. We performed additional experiments using DARPin-F5 without the 8xHis tag. We obtained very similar K_d values for the binding of DARPin-F5 with and without the 8xHis tag to LC/A1 using ITC measurements (Supplementary Figure S4). We included these results in the revised version of the manuscript.

2. It will be helpful if the authors can measure and compare the binding affinity for DARPin-F5 to full-length BoNT/A1 versus to LC/A1, as well as at different pH for LC/A1.

Reviewer 1 made a similar comment. We tried to determine the binding parameters of DARPin F5 to the full-length toxin by ITC, MST and SPR but despite numerous attempts we were not able to obtain conclusive results. We think that these complications are due to the belt that needs to be replaced by DARPin-F5.

However, in additional experiments, we show that DARPin-F5 binds to the toxin under neutral and acidic conditions. Notably, DARPin-F5 binding results in a destabilization of the toxin at both neutral and acidic conditions as determined by nanoDSF measurements (Supplementary Figure S6).

As requested by the reviewer, we measured the binding of DARPin-F5 to LC/A1 under neutral and acidic conditions by ITC. Comparison of the measurements demonstrates that the results are very similar (Supplementary Figure S4). We included the results in the revised version of the manuscript.

3. It will be helpful for authors to include the exact sequence of DARPin-F5 in either figure or supplementary figures, so others can evaluate the work and utilize this tool.

The sequence of DARPin F5 is included in the PDB file. However, we now also include the sequence of the original DARPin and the version in which the 8xHis tag can be removed by thrombin cleavage in Supplementary Figure 1.

Minor:

Figure 1: add a cartoon to illustrate the screening method and results.

A cartoon to illustrate the detailed screening method is published in Dreier and Plückthun (2012), *Methods Mol. Biol.* 261, which we cite (reference 41). We would prefer not to show it again because it is a standard procedure that has been performed many times for the selection of other DARPins. However, we now describe the method in detail in the Materials and Methods section of the revised version of the manuscript.

Figure 3c: it will be better to show the uncropped results here.

For clarity, we would prefer to show the cropped results. In the text, however, we now also refer to the Supplementary Figure with the uncropped gels in the text.

Abstract needs to be revised, start with a brief introduction to botulinum neurotoxins and to DARPins.

As a result of a limitation of words in the Abstract, we unfortunately can't add the requested introduction on botulinum neurotoxins and DARPins in the Abstract without removing important information on our findings.

If we are allowed to use excess words in the Abstract, we will add the following text, which contains the first sentence: "Botulinum neurotoxins (BoNTs), the causative agents of botulism are the most potent toxins. Despite their toxicity, they are the most widely used therapeutic proteins. In this study, we characterized Designed Ankyrin Repeat Proteins (DARPins) as investigative tools to probe BoNT function and as potential antidotes for botulism. DARPins are small engineered antibody mimetics derived from ankyrin proteins that are successfully used in various research, diagnostic and therapeutic applications."

Line 21: "Repeat"

Line 40: extra . before [5].

Line 30: in the United States

Line 109: needs full name for HTRF assay.

Line 221: add (CNGs)

Line 252: DARPIn-F5

Line 258: LC/A1

Line 335: delete this sentence.

Line 338: LC/A1

We addressed all these points as requested by the reviewer.

Reviewer #4

In the manuscript entitled "A DARPIn Increases the Catalytic Activity of Botulinum

Neurotoxin A1" Designed Ankyrin Repeat Proteins (DARPin)s are presented as investigative tools to probe botulinum neurotoxin (BoNT) function and as potential antidotes for botulism. One DARPin, DARPin-F5, was found to block SNAP25 cleavage by BoNT/A1 in vitro. X-ray crystallography revealed that DARPin-F5 inhibited BoNT/A1 activity by interacting with a SNAP25-binding region of the light chain metalloprotease between the α - and β -exosites. However, in contrast to these in vitro results, DARPin-F5 accelerated BoNT/A1 cleavage of the substrate in primary neurons. This result was confirmed in a mouse phrenic nerve hemidiaphragm assay, which showed faster paralysis in the presence of DARPin-F5.

The manuscript is well written and the data are convincing. Nonetheless, there are some weaknesses that need to be addressed.

First, the title of the manuscript mischaracterizes the findings. The results show that DARPin-F5 contributed to faster translocation of the BoNT/A1 light chain to the neuronal cytoplasm, and as a consequence, SNAP-25 cleavage was accelerated resulting in more rapid onset of BoNT-induced neuromuscular action. There is no evidence that DARPin-F5 contributed to increased enzymatic activity of BoNT/A1 LC metalloprotease. The correct description of the effect of the accelerated LC translocation is very often intertwined with incorrect statements related to "increased activity" (lines 31, 219, 225-227, 228, 243, 251, 272, 287-288, etc). There are "effects" of DARPins all of which are consequences of increased LC translocation. Therefore, all notes and sentences related to the "increased activity" should be revised accordingly.

Reviewer 1 made a similar comment concerning the title. We agree with both reviewers and changed the title to "A DARPin Promotes Faster Onset of Botulinum Neurotoxin A1 Action". We also revised all notes and sentences related to "increased activity" after we found that faster translocation is the mechanism for increased substrate cleavage.

Second, from the experimental evidence presented in the manuscript it is hard to envision how binding of DARPin-F5 to LC/A1 in the sub-nanomolar range contributes to accelerated relocation of the metalloprotease through the endosomal pore. This binding should contribute to stabilization of the LC/A1, similarly to the stabilization of the LC/A1 by nanobodies (lines 253-254 of the manuscript) and inability of the metalloprotease in complex with nanobodies to pass through endosomal pore to the neuronal cytoplasm. According to the results shown, the metalloprotease is not only able to translocate with higher efficiency, but also able to cleave the SNAP25 substrate in the neuronal cytosol, thus DARPin-F5 should remain in the endosome, i.e., should be spatially separated from the enzyme, otherwise LC/A1 would still in complex with DARPin, and SNAP25 cleavage would be blocked.

Examination of the data presented suggests an explanation for this contradiction, in that the interpretation of the experiment shown in Figure S2 may be a misinterpretation of the data. It is likely that DARPin-F5 binds the LC/A1 as a single entity and as a part of BoNT/A1 holotoxin at neutral/basic pH (otherwise it is hard to imagine how the DARPin not only

undergoes neuronal uptake with the toxin, but also generates an effect of faster translocation of metalloprotease reported in this manuscript). Thus, perhaps the shift of the protein peak in SEC (Figure 2S) toward higher MW under neutral pH and presence of DARPin-F5 in the fraction with the holotoxin shown in Figure S2 can be explained through an interaction of DARPin-F5 with LC/A1. However, the seemingly preserved interaction DARPin-F5 with LC/A1 at pH 5.2 may have a different explanation. The sequence of DARPin-F5 was not provided, thus it is not possible to determine the charge of this protein at pH 5.2. The electrostatic charge of BoNT/A1 holotoxin and DARPin-F5 at pH 5.2 may contribute to a relatively weak ionic interaction between these molecules; in the absence of high salt in the buffer used for SEC this interaction may be preserved and misinterpreted. It will be necessary to determine if DARPin-F5 still forms a high affinity complex with 1) LC/A1 and 2) with BoNT/A1 holotoxin at an acidic pH. This might be done using an experimental setting similar to that presented in Figure 4 and lines 203-217 of the manuscript.

If the DARPin-F5/LC/A1 interaction is absent/weakened at acidic pH, then observations from other experiments presented in the manuscript can be logically explained: 1) At neutral pH DARPin-F5 can bind both – LC/A1 and BoNT/A1; when it binds to holotoxin, the “belt” enveloping the LC/A1 probably becomes dislocated, as authors mentioned in the manuscript (line 252).

Reference 29 mentioned by the authors (lines 71, 72 of the manuscript) not only discusses “exposure of the hydrophobic peptide through a viral-fusion-peptide-like pH-dependent molecular switch,” but the role of the “belt” as an extended part of this switch, with two important statements: 1) “The belt of BoNT/A1 may modulate the conformational change of HN by interacting with and shielding the hydrophobic surface of the BoNT-switch. The belt is likely unfolded upon membrane binding and vesicle acidification that may help to release the BoNT-switch”; 2) “The HN core lacking the “belt” (termed tHN) was shown to form an ion channel independent of pH”.

The probable dislocation of the “belt” with DARPin-F5 (as authors proposed, line 252) at neutral pH exposes tHN, which, as mentioned above can form an ion channel independent of pH. Endosomal acidification would not only cause dissociation of DARPin-F5 and LC/A1 metalloprotease but would result in “globular melting” of the enzyme allowing it to pass through the narrow endosomal pore already formed before acidification by “uninhibited” tHN. In contrast, if, at an acidic endosomal pH, DARPin-F5 remains associated with LC/A1, then the effect would be similar to the prevention of translocation following stabilization of BoNT-LC by antibodies or nanobodies as mentioned in lines 253-254 and opposite to the observations described in this manuscript.

We thank the reviewer for this very detailed analysis and possible explanation of our results. Motivated by this, we performed several additional experiments.

First, we analysed the binding of DARPin-F5 to LC/A1 at neutral and acidic conditions. We found that DARPin-F5 bound with a similar affinity to LC/A1 at acidic and neutral conditions

(Supplementary Figure S4). Next, we analyzed the stability of the full-length toxin with and without DARPin F5 under neutral and acidic conditions by nanoDSF (Supplementary Figure S6). We found that DARPin-F5 destabilized the full-length toxin, which would be consistent with the well-accepted view that destabilization of the toxin could result in faster translocation of the LC. Based on our new findings, we revised the Discussion section of the manuscript. Furthermore, we describe the role of the belt in the translocation process as outlined by the reviewer.

The possibility that dislocation of the belt by DARPin-F5 has an impact on translocation is a very clever suggestion. However, it is difficult to believe that the H_N core lacking the “belt” (termed tHN) will form an ion channel independent of pH in the context of the full-length toxin. To our knowledge, ion channel formation and insertion is not pH-independent. It is modulated by H_C as described by Montal and colleagues (Fischer and Montal, *Toxicon*, 2013).

It should also be mentioned that there exist results published in the literature that cannot simply be explained by stabilization/destabilization of the toxin. While Binz and co-workers (Bade et al., *J. Neurochem*, 2004) showed that translocation depends on the stability of proteins fused to the N-terminus of BoNT/D, two recent reports (Miyashita et al., *Sci Transl. Med.*, 2021; NcNutt et al., *Sci Transl. Med.*, 2021) demonstrated that inactive BoNT/C1 or an inactive chimera between BoNT/X and BoNTA can serve as vehicles to translocate single-chain antibodies targeting LC/A1. Typically, single-chain antibodies are very stable proteins and therefore should prevent translocation. However, in the two mentioned reports the single-chain antibodies were successfully translocated into the neuronal cytoplasm where they inhibited SNAP-25 cleavage. If we assume that the LC of BoNTs is pulled through the translocation pore in a manner similar to the one that has been suggested for diphtheria toxin (Murphy, *Toxins*, 2011), then controversial findings could possibly be explained by differences between mechanical and thermal LC stability. To my knowledge, this issue has not been addressed.

Minor issues:

Line 21 – Repeat, not Repat.

Corrected.

Line 121 – What is the percent inhibition when the molar ratio is 1:1?

At least 80%

Line 172 – Interesting that all other LC/A subtypes have K128 and D131 preserved. That means a single amino acid – E171 is determinative for the high affinity interaction between DARPin-F5 and LC/A1. Just an absence of a single methylene group in the side-chain (E>D mutation) is able to abolish this binding.

Subtype BoNT/A6 has also a Glu residue at position 171 and therefore is very likely to bind DARPin-F5. Mutation of Glu-171 to Asp doesn't completely abolish binding but reduces it >5000 fold.

Lines 175-181 – is this explanation really necessary?

Based on the comments of reviewers 2 and 3, we can't remove the explanation.

Line 258 – to compare, not to estimate.

We replaced "estimate" by "compare".

Lines 259, 261, and elsewhere – how long is the preincubation? Is it necessary to "preincubate" instead of just "mix" the components?

Because DARPin-F5 needs to dislocate the belt before binding to BoNT/A1, we typically incubated the samples for at least 1h. We added this information to the main text of the manuscript.

Lines 298, 299 "with a recent publication." The citation to Cai et al is from 2006. A 17-year-old paper is not recent.

We removed "recent".

Lines 308-321 – seems not really relevant to the observations reported in the manuscript.

Because the conformation of the toxin (compact or closed) is known to have an effect on translocation efficiency (Kumaran et al., J. Mol. Biol., 2009), we used this paragraph as a possible explanation of what might be the result of DARPin-F5 binding.

Please note that it also very well complements the new paragraph on faster acting toxins.

Lines 325 – 335 – faster acting toxin and lower dose of the toxin – are totally different characteristics. By using a faster acting toxin, the onset is faster but the actual dose of the toxin supposed to be the same (unless of course the slow-acting toxin instead of translocating to the cytosol get degraded through late endosome and lysosome), therefore this entire paragraph needs to be rewritten.

We thank the reviewer very much for this comment. Our findings indeed show faster onset of toxin action. Therefore we completely changed the paragraph and now focus on a comparison to BoNT/E and F and the benefits of faster acting toxins for certain applications.

The formation of an immune response to the toxin is irrelevant because of what was said above. By using DARPin-F5 as a means to provide faster onset may result in formation of

immune response to the DARPIn-F5 itself (which is used in significant (25 times) molar excess in this manuscript).

This point was addressed by Andreas Plückthun as follows. DARPins have been in clinical trials for systemic use in human patients with multiple injections over prolonged times. The non-randomized backbone is free of T-cell epitopes and among the tight and specific binders, those can be chosen which do not carry T-cell epitopes in the randomized region either. Since DARPins are very resistant to aggregation, they do not induce a T-cell-independent antibody response.

In summary, DARPins can be used in systemic injections in humans.

Figure S2 – What are the additional protein bands running below un-reduced holotoxin at pH 5.2? Aren't these bands the consequence of unexpected reduction?

From their sizes, the bands are not corresponding to the heavy chain and the light chain of BoNT/A1. To demonstrate that, we included the positions of size markers. The most likely explanation is some degradation under acidic conditions.

Reviewers' Comments:

Reviewer #1:

Remarks to the Author:

All my concerns have been adequately addressed by the authors.

Reviewer #2:

Remarks to the Author:

The revised manuscript includes new DSF results to show the destabilization afforded by the Darpin at low pH and the K_i measurements with His tag removed, better supporting the authors main hypothesis and the physiological significance of the crystal structure. The authors have addressed the majority of my concerns. One point though is that the separate B-factors for BoNT, Darpin and solvent, although provided in the response to reviewers, needs to be added to Table S1, as I stated in the review of the original manuscript. This is for the benefit of the general reader and the more expert reader so that they do not need to analyze the molecule themselves from the PDB.

Reviewer #3:

Remarks to the Author:

The authors' revision is satisfactory. It is intriguing that authors have difficulty to obtain data on binding of DARPIn-F5 to full-length toxin by ITC, MST and SPR. It will be helpful to briefly discuss this point and possible reasons in the Discussion section.

Reviewer #4:

Remarks to the Author:

Although the authors have improved the manuscript considerably, there are several important issues remaining that must be addressed:

1. In response to the criticism that "increased activity" was an incorrect manner of characterizing accelerated LC translocation, the authors stated that "we revised all notes and sentences related to "increased activity" after we found that faster translocation is the mechanism for increased substrate cleavage". However, in the current revision they still use "increased activity" (lines 237, 243, 246 of the manuscript and possibly elsewhere in the text), which is inappropriate. They could perhaps use the term "rate of substrate / SNAP-25 cleavage," rather than "increased activity".

2. By providing additional experimental data, the authors have addressed most of the reviewer criticisms; however, I believe that they have introduced additional ambiguity. For example: a) on Figure 4, panel (b) the K_D of LC/A1 interaction with DARPIn-F5 as determined by surface plasmon resonance is 2.38×10^{-10} M; on Supplementary Figure S4, panel (a), where parameters of LC/A1 interaction with DARPIn-F5 were evaluated by isothermal titration calorimetry, the K_D for the same pair was determined to be $9.61 \times 10^{-8} \pm 0.02$ M. The difference between these numbers is almost three orders of magnitude. Which number is correct? Do they have an explanation for the difference?; b) Supplementary Figure S6, in which relative stability of BoNT/A1 alone and in complex with DARPIn-F5 was evaluated at pH 5.5 and 7.4 by nanoscale differential scanning fluorimetry, indeed, shows a destabilization effect of DARPIn-F5 on the complex at both pH values – acidic (panel a, pH 5.5) and slightly basic (panel b, pH 7.4). The degree of destabilization is more pronounced under acidic conditions ($\Delta T = \sim 1.6^\circ\text{C}$) than at slightly basic ($\Delta T = \sim 0.9^\circ\text{C}$). However, the absolute values of temperatures when 50% of the proteins are supposed to be in the "melted" state at either pH are questionable. For instance, the value for BoNT/A1 alone at pH 7.4 is about 51.9°C versus 53.7°C at pH 5.5, suggesting that at the acidic pH required for LC/A1 unfolding, the BoNT/A1 holotoxin is more stable. Is there any explanation for these numbers?

3. The evidence indicating a DARPIn-F5 interaction with BoNT/A1 at neutral and acidic pH shown by the shift of the peak on SEC (Supplementary Figure S3) is fairly weak; data shown in Supplementary Figure S4 for a DARPIn-F5 interaction with LC/A1 at pH 7.5 and 5.5 suggest that

the acidic pH does not contribute to strengthening of this interaction (K_D $7.01 \times 10^{-8} \pm 0.01$ at pH 5.5 vs K_D $4.43 \times 10^{-8} \pm 0.01$ at pH 7.5). The same should be true for the interaction of DARPin-F5 with BoNT/A1 holotoxin, in which DARPin-F5 competes with the belt region for binding to LC/A1 contributing to the displacement of the belt resulting in easier formation of translocation pore. However, as a result of this competition with the belt, the DARPin-F5 interaction with LC/A1 in the BoNT/A1 holotoxin should be weakened in comparison with stand alone LC/A1. In response to the first comment of reviewer 1, the authors wrote: "We tried to determine the binding parameters of DARPin-F5 to the full-length toxin by ITC, MST and SPR but despite numerous attempts, we were not able to obtain conclusive results. We think that these complications are due to presence of the belt that as indicated by the reviewer needs to be replaced before DARPin-F5 can bind to the LC." It is not clear why slow kinetics of interaction were not detected by SPR or ITC; however, the authors were able to show the strength of this interaction at different pH through evaluation of the thermal stability of the BoNT/A1-DARPin-F5 complex (Supplementary Figure S6). As an alternative means for assessing interaction strength by SPR or ITC, the authors could and should examine the thermal stability of LC/A1 and LC/A1-DARPin-F5 complex by nanoscale differential scanning fluorimetry. This experiment will provide an answer regarding the relative strength of the DARPin-F5 interaction with BoNT/A1 holotoxin vs the interaction of DARPin-F5 with stand-alone LC/A1, and can perhaps identify the driving factors behind separation of DARPin-F5 from the light chain of the BoNT/A1 holotoxin at endosomal pH.

4. Minor issues:

line 204 – assessed, not assessed

line 264 – clearly, not clearly

line 356 – successfully, not successfully

Supplementary Figure S5 and legend – what is DARPin 18? Is it in reality DARPin-F5?

POINT-BY-POINT RESPONSE TO REVIEWERS' COMMENTS

Reviewer #1

All my concerns have been adequately addressed by the authors.

Reviewer #2

The revised manuscript includes new DSF results to show the destabilization afforded by the Darpin at low pH and the K_i measurements with His tag removed, better supporting the authors main hypothesis and the physiological significance of the crystal structure. The authors have addressed the majority of my concerns. One point though is that the separate B-factors for BoNT, Darpin and solvent, although provided in the response to reviewers, needs to be added to Table S1, as I stated in the review of the original manuscript. This is for the benefit of the general reader and the more expert reader so that they do not need to analyze the molecule themselves from the PDB.

We apologize for this careless mistake. We now include the separate B-factors for LC/A1, DARPin-F5 and solvent in Supplementary Table S1.

Reviewer #3

The authors' revision is satisfactory. It is intriguing that authors have difficulty to obtain data on binding of DARPin-F5 to full-length toxin by ITC, MST and SPR. It will be helpful to briefly discuss this point and possible reasons in the Discussion section.

As suggested by the reviewer, we included a short paragraph describing our difficulties in obtaining data on the binding of DARPin-F5 to full-length toxin by ITC, MST and SPR and the possible reasons on page 11 of the Results section. We feel that this is the better place than the Discussion section.

We were not able to obtain data on the binding of DARPin-F5 to full-length toxin by several methods, which is most likely mainly the result of a combination of DARPin-F5 binding and belt dislocation, which might also cause conformational changes in the toxin. Our ITC data look like a mixture of endothermic high-affinity and exothermic low affinity titrations. However, the quality of the data was never good enough for proper fitting. For MST, we obtained binding data that were impossible to interpret, because the bound/unbound differences change depending on the incubation time. Furthermore, we never reached a plateau of the binding curve at high concentrations, which is necessary for K_d evaluation. For SPR, we observed unspecific binding of DARPin-F5 to the chip matrix that masked a potential signal.

Reviewer #4

Although the authors have improved the manuscript considerably, there are several important issues remaining that must be addressed:

1. In response to the criticism that "increased activity" was an incorrect manner of characterizing accelerated LC translocation, the authors stated that "we revised all notes and sentences related to "increased activity" after we found that faster translocation is the mechanism for increased substrate cleavage". However, in the current revision they still use "increased activity" (lines 237, 243, 246 of

the manuscript and possibly elsewhere in the text), which is inappropriate. They could perhaps use the term “rate of substrate / SNAP-25 cleavage,” rather than “increased activity”.

As suggested by the reviewer, we replaced “increased activity” by “increased rate of substrate / SNAP-25 cleavage”.

2. By providing additional experimental data, the authors have addressed most of the reviewer criticisms; however, I believe that they have introduced additional ambiguity. For example: a) on Figure 4, panel (b) the KD of LC/A1 interaction with DARPIn-F5 as determined by surface plasmon resonance is 2.38×10^{-10} M; on Supplementary Figure S4, panel (a), where parameters of LC/A1 interaction with DARPIn-F5 were evaluated by isothermal titration calorimetry, the KD for the same pair was determined to be $9.61 \times 10^{-8} \pm 0.02$ M. The difference between these numbers is almost three orders of magnitude. Which number is correct? Do they have an explanation for the difference?

The ITC run looks good and reasonable. An N-value of ~ 1.1 shows that the concentrations are reasonably correct and the $\frac{3}{4}$ points in the slope are sufficient for a good fit.

The SPR run might overestimate the on rate because it is close to diffusion limit at the apparent immobilization level. The off rate might be underestimated because we never reached a plateau. If both of those rates are off by one order of magnitude one would get into a similar range as the ITC. As the steady state was not reached it is difficult to deconvolute this. However, we repeated the experiment also on our OpenSPR instrument and obtained results that are in good agreement to the SPR data. Furthermore, our experiments in cells and in muscle tissue, that were conducted at very low toxin concentration, also demonstrate that the binding of DARPIn-F5 is very strong.

For these reasons, it is impossible to decide which value is the correct one.

b) Supplementary Figure S6, in which relative stability of BoNT/A1 alone and in complex with DARPIn-F5 was evaluated at pH 5.5 and 7.4 by nanoscale differential scanning fluorimetry, indeed, shows a destabilization effect of DARPIn-F5 on the complex at both pH values – acidic (panel a, pH 5.5) and slightly basic (panel b, pH 7.4). The degree of destabilization is more pronounced under acidic conditions ($\Delta T_m = \sim 1.6^\circ\text{C}$) than at slightly basic ($\Delta T_m = \sim 0.9^\circ\text{C}$). However, the absolute values of temperatures when 50% of the proteins are supposed to be in the “melted” state at either pH are questionable. For instance, the value for BoNT/A1 alone at pH 7.4 is about 51.9°C versus 53.7°C at pH 5.5, suggesting that at the acidic pH required for LC/A1 unfolding, the BoNT/A1 holotoxin is more stable. Is there any explanation for these numbers?

As the reviewer knows, pH changes to more acidic values and the local micro-environment affect primarily His and negatively charged residues. Protonation of these residues can influence their packing and result in increased or decreased electrostatic repulsion, which might affect salt bridge formation. The sum of these stabilizing and destabilizing effects that affects all three BoNT domains result in the observed stability (in this case thermal stability) differences compared to the toxin at physiological pH. While it is generally accepted that translocation requires acidic conditions, work remains to be done to clarify whether thermal or mechanical protein stability is the force relevant for LC translocation.

3. The evidence indicating a DARPIn-F5 interaction with BoNT/A1 at neutral and acidic pH shown by the shift of the peak on SEC (Supplementary Figure S3) is fairly weak; data shown in Supplementary

Figure S4 for a DARPin-F5 interaction with LC/A1 at pH 7.5 and 5.5 suggest that the acidic pH does not contribute to strengthening of this interaction ($KD 7.01 \times 10^{-8} \pm 0.01$ at pH 5.5 vs $KD 4.43 \times 10^{-8} \pm 0.01$ at pH 7.5). The same should be true for the interaction of DARPin-F5 with BoNT/A1 holotoxin, in which DARPin-F5 competes with the belt region for binding to LC/A1 contributing to the displacement of the belt resulting in easier formation of translocation pore. However, as a result of this competition with the belt, the DARPin-F5 interaction with LC/A1 in the BoNT/A1 holotoxin should be weakened in comparison with stand alone LC/A1. In response to the first comment of reviewer 1, the authors wrote: “We tried to determine the binding parameters of DARPin-F5 to the full-length toxin by ITC, MST and SPR but despite numerous attempts, we were not able to obtain conclusive results. We think that these complications are due to presence of the belt that as indicated by the reviewer needs to be replaced before DARPin-F5 can bind to the LC.” It is not clear why slow kinetics of interaction were not detected by SPR or ITC; however, the authors were able to show the strength of this interaction at different pH through evaluation of the thermal stability of the BoNT/A1-DARPin-F5 complex (Supplementary Figure S6). As an alternative means for assessing interaction strength by SPR or ITC, the authors could and should examine the thermal stability of LC/A1 and LC/A1-DARPin-F5 complex by nanoscale differential scanning fluorimetry. This experiment will provide an answer regarding the relative strength of the DARPin-F5 interaction with BoNT/A1 holotoxin vs the interaction of DARPin-F5 with stand-alone LC/A1 and can perhaps identify the driving factors behind separation of DARPin-F5 from the light chain of the BoNT/A1 holotoxin at endosomal pH.

As suggested by the reviewer, we performed additional nanoDSF experiments with LC/A1 with and without DARPin-F5 at physiological and acidic pH. Notably, the unfolding transition of LC/A1 alone occurs over a much broader temperature range compared to the one of the full-length toxin. This finding might reflect the presence of an ensemble of LC/A1 structures that was proposed to exist under physiological conditions [50]. It is also consistent the presence of a molten golbule-like structure of the catalytic domain at acidic pH, which is not observed in the context of the full-length toxin [51]. DARPin-F5 destabilizes LC/A1 at physiological pH but stabilizes the catalytic domain at acidic pH. These results suggest that destabilization of BoNT/A1 is primarily caused by the dislocation of the belt by DARPin-F5. This might also affect the translocation domain, a conclusion that is consistent with a recent statement Lam and colleagues (Lam et al. (2020) Toxins), who studied single-chain camelid antibodies that block translocation of BoNT/E1. They write “Since H_N is suggested to be responsible for the rapid onset of BoNT/E intoxication [10], we hypothesize that the exceptional flexibility of the belt may contribute to the speedy translocation of BoNT/E1 by lowering the energy requirement for protein unfolding and LC delivery across the H_N channel. Therefore, testing the physiological role of the belt in facilitating BoNT/E translocation is of high interest in future studies.”

We included these results in the revised version of the manuscript on pages 12 and 14 and Supplementary Figure S6.

4. Minor issues:

line 204 – assessed, not assesed

line 264 – clearly, not cleary

line 356 – successfully, not sucessfully

Supplementary Figure S5 and legend – what is DARPin 18? Is it in reality DARPin-F5?

We apologize for these careless mistakes. They were corrected in the revised version of the manuscript.

Reviewers' Comments:

Reviewer #4:

Remarks to the Author:

Revised Supplementary Figure 6 provides additional logic to the manuscript. The authors answered all my raised questions and concerns.